# TRACE LENGTH IS A SIMPLE UNCERTAINTY SIGNAL IN REASONING MODELS

## ABSTRACT

Uncertainty quantification for LLMs is a key research direction towards addressing hallucination and other issues that limit their reliable deployment. In this work, we show that *reasoning trace length* is a simple and useful confidence estimator in large reasoning models. Through comprehensive experiments across multiple models, datasets, and prompts, we show that trace length performs in comparable but complementary ways to other zero-shot confidence estimators such as verbalized confidence. Our work reveals that reasoning post-training fundamentally alters the relationship between trace length and accuracy, going beyond prior work that had shown that post-training causes traces to grow longer in general (e.g., "overthinking"). We investigate the mechanisms behind trace length's performance as a confidence signal, observing that the effect remains even after adjusting for confounders such as problem difficulty and GRPO-induced length bias. We identify high-entropy or "forking" tokens as playing a key role in the mechanism. Our findings demonstrate that reasoning post-training enhances uncertainty quantification beyond verbal expressions, and establish trace length as a practical confidence measure for large reasoning models.

## 1 INTRODUCTION

As large language models (LLMs) demonstrate increasingly sophisticated capabilities, issues of hallucination and factual inaccuracy remain a barrier to their reliable and ethical widespread deployment (Kalai et al., 2025; Huang et al., 2025). One promising approach to addressing these limitations is to augment models with confidence estimates that quantify their uncertainty (Xiong et al., 2024; Tian et al., 2023). Such confidence measures can help users determine when to trust LLM outputs and when to exercise skepticism (Vodrahalli et al., 2022; Srinivasan & Thomason, 2025; Donahue et al., 2022).

There have been a variety of approaches proposed in the literature on LLM uncertainty quantification (LLM UQ; see Liu et al. (2025b) for a survey). The most practically useful methods are those that work in a *zero-shot* manner and require no training or finetuning, and can work with only a single generated sample per query. Arguably, the most straightforward approach that falls into this paradigm is verbalized confidence estimation, which simply asks the LLM for its confidence as it answers a particular question (Lin et al., 2022; Xiong et al., 2024). Verbalized confidence elicitation has two key beneficial properties: (1) it can be applied to black-box models (albeit with prompt modifications; see Appendix E); and (2) it is highly efficient. This eliminates the need to re-sample or aggregate multiple LLM responses, a requirement for other black-box UQ methods like semantic entropy (Kuhn et al., 2023).

With the advent of large reasoning models (LRMs) — LLMs post-trained either directly with RL algorithms like GRPO on large datasets of mathematical and scientific reasoning problems, or with fine-tuning on such models' reasoning traces — there has been a renewed interest in verbalized confidence methods for uncertainty quantification (Zeng et al., 2025; Mei et al., 2025). Recent work by Yoon et al. (2025) provides empirical evidence that LRMs have more calibrated verbalized uncertainty estimates when compared to their equivalent pre-RL variants. Zhang et al. (2025a) show that linear probes on LRM hidden states trained to predict the correctness of the output can be used to improve the token efficiency of generations via early termination.

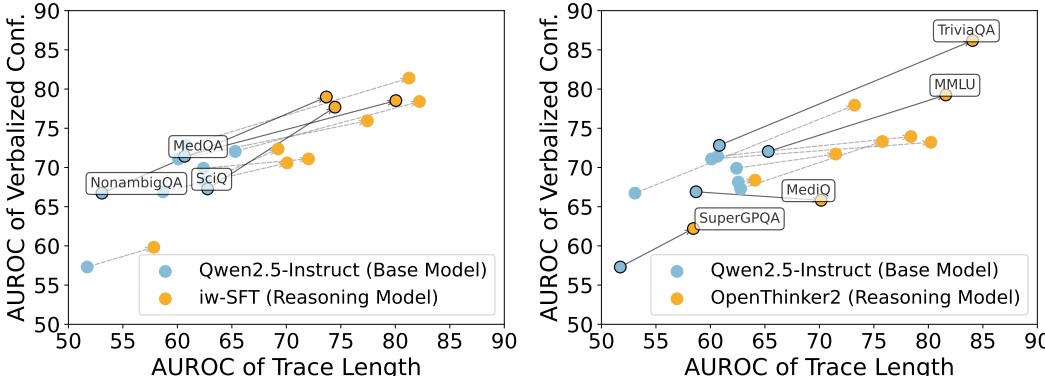

Figure 1: **Reasoning post-training improves both verbalized confidence and trace length as uncertainty signals.** Scatter plot showing verbalized confidence and trace length performance in terms of AUROC. Each point represents a different dataset, and arrows connect the same dataset before and after reasoning post-training. Two reasoning models — iw-SFT-32B (Qin & Springenberg, 2025) and OpenThinker2-32B (Guha et al., 2025) — have better verbalized confidence performance than their base model Qwen2.5-32B-Instruct for many datasets (utilizing Prompt 2). However, the *trace length* also emerges as a powerful uncertainty signal after post-training, and has comparable power in predicting whether the response was correct.

These findings suggest that reasoning post-training may fundamentally alter how models understand and express their own uncertainty. However, several important questions remain underexplored: How robust are these improvements in confidence estimation across different settings? Do the benefits of reasoning training extend beyond verbalized confidence to other forms of zero-shot uncertainty expression? And what mechanisms drive these improvements?

In this work we study an alternative confidence signal: the *length* of reasoning traces themselves. Intuitively, if a model is uncertain about a problem, it might engage in more extensive reasoning, producing longer traces as it works through its uncertainty. Problems the model finds straightforward might elicit shorter, more direct responses. While prior work has shown that reasoning post-training causes trace lengths to grow in general (Yi et al., 2025; Shen et al., 2025; Guha et al., 2025; Jin et al., 2025), its utility as a zero-shot confidence signal and how this might be affected by reasoning training remains largely unexplored.

We investigate these questions through comprehensive experiments across multiple models, datasets, and prompting strategies. Our main contributions are as follows.

1. **Trace Length as an Emergent Zero-Shot Confidence Estimate.** In Section 4, we show that reasoning trace length becomes a meaningful zero-shot confidence estimate after reasoning post-training, performing comparably to verbalized confidence across a range of settings (see Figure 1). Crucially, this signal is not present in base models, suggesting that the post-training process fundamentally alters the relationship between trace length and correctness. Furthermore, unlike verbal confidence elicitation, it requires no prompt modification, making it especially suitable for black-box use at inference time.

2. **Comparative Analysis of Verbal Confidence and Trace Length.** In Section 4.1, we analyze the relationship between verbal confidence and trace length, finding that while these measures become correlated after reasoning training, they capture complementary aspects of uncertainty. We show that simple combinations of both signals yield superior confidence estimates compared to either measure alone.

3. **Mechanisms Behind Trace Length's Emergence as a Confidence Estimate.** In Section 5, we investigate the question of why trace length becomes a reliable confidence signal after reasoning training, observing that the effect remains even after controlling for potential confounders such as problem difficulty and GRPO-induced length bias. We identify high-entropy or "forking" tokens (which often coincide with intuitive epistemic markers such as "maybe" and "wait") as playing a key role. We show that the number of forking tokens in a trace is another simple zero-shot confidence measure that only emerges after reasoning post-training, and is highly correlated with

trace length. Moreover, the average entropy of a token is directly predictive of its usefulness as an overall uncertainty indicator. Remarkably, even counting the occurrences of the single highest-entropy token within a trace achieves strong UQ performance. We believe that a deeper study of forking tokens will be a key step towards the scientific understanding of uncertainty in LLMs.

## 2 RELATED WORK

We cover some of the most closely related work here and defer additional work to Appendix A.

**Trace Length in Reasoning Models.** Recent research has examined how reasoning training affects trace length in language models. Dimakis (2025) observed that reasoning models, particularly Deepseek-R1, generate longer reasoning traces for incorrect answers than correct ones, a pattern subsequently confirmed by other studies (Marjanović et al., 2025; Balachandran et al., 2025; Shrivastava et al., 2025; Ballon et al., 2025). Notably, Zheng et al. (2023) observe that instruction-tuned LLMs can often accurately predict the length of their own answers. While prior work has utilized this observation to develop methods for reducing trace length or improving accuracy (Qu et al., 2025; Shrivastava et al., 2025; Hassid et al., 2025; Wu et al., 2025), we explore trace length as an indicator of model confidence. Beyond the documented difference in mean lengths, we demonstrate that trace length provides a fine-grained confidence signal capable of distinguishing between likely-correct and likely-incorrect responses. Most closely related to our work is Vanhoyweghen et al. (2025), who examine various heuristics derived from reasoning traces to predict response accuracy. However, their evaluation is limited to two models and two datasets, one of which is exceptionally challenging (achieving sub-10% model accuracy). In contrast, our evaluation spans a diverse range of datasets with varying difficulty levels, revealing that trace length serves as a substantially stronger indicator of correctness than suggested by the findings of Vanhoyweghen et al. (2025).

**Post-Training-Free Confidence Estimates in Reasoning Models.** We provide a brief overview of existing techniques for confidence estimation in language models that do not require direct fine-tuning. Several recent works have evaluated LLMs' self-verbalized confidence as a zero-shot confidence estimation approach (Zeng et al., 2025; Yoon et al., 2025), finding that it often performs reasonably well and shows improvement following reasoning training. Beyond verbalized confidence, existing work explores alternative UQ techniques that require varying degrees of model access. These include methods that leverage internal token probabilities (Duan et al., 2024), approaches that train probes to predict uncertainty based on internal model states (Kossen et al., 2024; Zhang et al., 2025a), and techniques that utilize multiple generations from the model's predictive distribution for each query (Kuhn et al., 2023; Manakul et al., 2023; Farquhar et al., 2024). In a comprehensive comparison of these methods, Tao et al. (2025) found that verbalized confidence estimates generally outperform other single-pass black-box LLM UQ methods.

## 3 EXPERIMENTAL SETUP: DATASETS, MODELS, AND EVALUATION

**Models.** We evaluate four 32B reasoning models: iw-SFT (Qin & Springenberg, 2025), Open-Thinker2 (Guha et al., 2025), Skywork-OR1 (He et al., 2025a), and R1-Distill (Guo et al., 2025). Importantly, each model is a fine-tuned variant of Qwen2.5-32B (Yang et al., 2024a). Similar to Yoon et al. (2025), this allows us to quantify how various post-training approaches influence the verbalized confidence abilities of the resulting model. We also evaluate two 7B reasoning models based on Qwen2.5-7B-Instruct: Nemotron-7B (Nathawani et al., 2025) and OpenThinker3-7B. Details on models are in Appendix C. We point out that iw-SFT and OpenThinker are versions of Qwen2.5-32B-Instruct which are supervised fine-tuned (SFT) with completely *open* data and code. As such, we have complete knowledge of the post-training procedure, and guessing whether certain additional post-training steps were taken which could have improved or reduced the performance of verbalized confidence estimation abilities is not required. In contrast, we note that R1-Distill is a fine-tuned version of the (non-instruct) Qwen2.5-32B, the data used to create R1-Distill is private, and Skyworks-OR1 uses R1-Distill as its base model.

**Prompts and Evaluation.** We run our evaluations using three standard verbalized confidence prompts detailed in Appendix E: (1) a *linguistic* confidence prompt which asks a model to output confidence phrases such as "Highly Likely" (Prompt 1, taken from Yoon et al. (2025)); (2) a *numeric* confidence elicited from the range $[0, 100]$ (Prompt 2, taken from Mei et al. (2025)); and

(3) A top-$k$ prompt for $k = 5$ (Prompt 3, from Mei et al. (2025); Tian et al. (2023)). We also use the identical evaluation framework of Yoon et al. (2025); namely, we use evalchemy (Raoof et al., 2025) as well as lm-evaluation-harness (Gao et al., 2024) for efficient inference with vLLM.

**Datasets.** We report results on ten datasets detailed and cited in Appendix D. The datasets span simple mathematical (MMLU) and non-mathematical reasoning (MMLU-Pro-NoMath), datasets built to measure aleatoric uncertainty (FolkTexts), and multiple choice / free-response QA questions (TriviaQA, NonambigQA, MedQA, etc.).

## 3.1 Evaluation Metrics

The standard approach for evaluating confidence expressions (Yoon et al., 2025; Xiong et al., 2024) often involves reporting multiple metrics: accuracy, Brier score, Expected Calibration Error (ECE), and Area Under the Receiver Operating Characteristic curve (AUROC), to name a few. However, these metrics can often tell conflicting stories about the performance of verbal confidence in a model; in Appendix B, we provide an example where a lower ECE is not accompanied by a higher AUROC.

For zero-shot confidence estimates specifically, especially verbalized confidence, we view AUROC as the most useful metric to study. AUROC has the following properties that make it suitable for UQ in real-world applications involving LLMs:

1. **Captures discriminative ability.** A useful metric must reward an uncertainty measure for distinguishing meaningfully between less likely and more likely predictions, and between correct and incorrect predictions. Both AUROC and Brier score do this. By contrast, ECE measures only whether confidence levels align with empirical accuracy within predefined bins, ignoring whether the model can meaningfully differentiate between high and low confidence cases. This can lead ECE to paradoxically reward uninformative estimates while penalizing genuinely useful ones. For example, zero ECE can be achieved by uniformly expressing 70% confidence, say, on all inputs simply because the average accuracy is 70%.
2. **Insensitive to nominal values.** AUROC is a purely rank-based metric for uncertainty, and is insensitive to the exact values that the UQ takes. This makes it well-suited to evaluating confidence estimates that don't naturally output precise probabilities, such as verbalized confidence approaches where models are asked to express uncertainty through linguistic phrases. By contrast, value-based metrics such as Brier score and ECE may vary significantly based on superficial aspects of the evaluation setup, such as whether "very likely" is interpreted as corresponding to 0.90, 0.95, or 0.99.

We include a more detailed discussion of these issues — including illustrative pathological cases demonstrated by reasoning models — in Appendix B. Throughout, we will report AUROC $\times 100$ for better readability (e.g., $0.75 \rightarrow 75$). We also note that AUROC can be computed for any collection $\{(x_i, y_i, s_i)\}_i$, where $(x_i, y_i)$ is the $i$th labeled example, and $s_i \in \mathbb{R}$ an associated "uncertainty score" for that example (for instance, the numeric verbal confidence). Unlike most calibration metrics, computing AUROC involves no binning or normalization of scores (see Appendix B.2).

## 4 Reasoning Trace Length is an Emergent Confidence Signal

In this section, we evaluate reasoning trace length as a simple zero-shot uncertainty measure, comparing to verbalized confidence as a baseline. We first show that both trace length and verbalized confidence become useful only after reasoning post-training, although the magnitude of improvement over the base non-reasoning models varies by prompt and dataset. Then, we explore connections between the two methods, including examining how correlated they are, and a simple way to combine them for (nearly) strictly better UQ.

To start, in Figure 2 we demonstrate that the average performance of *trace length* (TL) over all models, prompts, and datasets is comparable with that of verbalized confidence (VC). In particular, TL is within 2-3 points of VC in AUROC[1] across all models. Figure 2 also confirms the positive findings of the verbalized UQ abilities improving over the base Qwen2.5 models after reasoning

---

[1] Note that to compute the AUROC of trace length, we take the score $s_i \in \mathbb{R}$ to be the *negative* generation length in characters, as longer trace length implies that model is less confident. AUROC is then computed in the standard manner (see Appendix B).

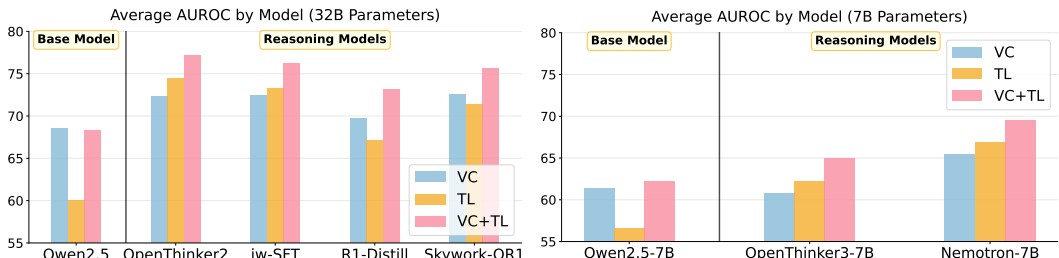

Figure 2: **AUROC performance of verbalized confidence (VC), trace length (TL), and their zero-shot sum VC+TL.** (**Left**): Performance for the 32B base model Qwen2.5-Instruct and four 32B reasoning models post-trained from Qwen2.5 (see Appendix C for model details). Results are averaged over ten datasets (Appendix D) and three prompts (Appendix E) per model. After reasoning post-training: (1) trace length emerges as a reliable uncertainty signal, competitive with verbal confidence; and (2) summing VC and TL together almost always outperforms both TL and VC individually. (**Right**): We find similar results for two 7B reasoning models post-trained from Qwen2.5-7B.

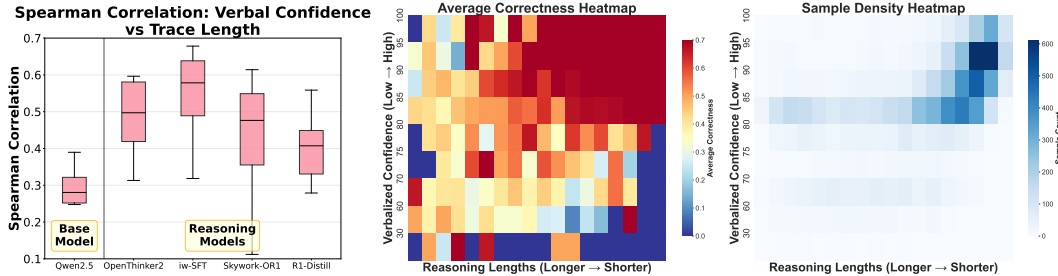

Figure 3: **Verbalized confidence (VC) and Trace Length (TL) are only loosely correlated.** (**Left**): Distribution over ten datasets of spearman correlations between VC and TL per 32B model, demonstrating that the two quantities are correlated but not perfectly so (using Prompt 1). (**Middle & Right**): Heatmap of average correctness (middle) and sample density (right) for OpenThinker2-32B using Prompt 2 over all datasets, split by VC and TL. The upper right quadrant of the center heatmap demonstrates that using only VC or TL as an uncertainty measure in isolation (e.g., choosing a horizontal or vertical threshold) will not outperform using VC + TL.

post-training (Yoon et al., 2025). When considering the average performance over all datasets and prompts, verbalized confidence provides improved utility for all but one of the reasoning models considered (OpenThinker3-7B is the exception). Full tables for each model, dataset, and prompt are in Appendix F.1 for 32B models and F.2 for 7B models.

We stress that trace length is still a useful UQ tool even if the prompt does not ask the model to explicitly reason about its confidence. In Appendix F.3, we show that the AUROC of trace length remains stable even using a vanilla prompt that asks the model only to reason about its answer, and not its confidence (see Prompt 4 for the standard answer-only prompt used). This means that we can often match the performance of verbalized confidence estimation without any prompt modification at all. More broadly, we propose that trace length should be considered as a standard baseline in zero-shot UQ in reasoning models, since it can be collected and measured at no additional cost or prompt modifications.

### 4.1 CONNECTIONS BETWEEN TRACE LENGTH AND VERBALIZED UNCERTAINTY

To what extent are verbalized uncertainty and trace length capturing the same signals of uncertainty from the model? In other words, to what extent can we improve our uncertainty estimates by relying on a combination of length and verbalized uncertainty, rather than just length alone?

To investigate this question, we first consider the correlation between the AUROC of verbal confidence and the AUROC of the trace length across different models and datasets (Figure 3, left).

With the exception of one model (R1-Distill), we observe a fairly strong correlation between the performance of verbal confidence and trace length as uncertainty measures across different datasets. This correlation suggests that the characteristics of data and model training that encourage self-verbalized confidence to be a valuable confidence estimate may also be helping length itself become a good uncertainty estimate.

In the middle of Figure 3, we show a heatmap of the correctness of OpenThinker given particular reasoning length and verbalized confidence values. The heatmap demonstrates that knowing the verbal confidence (horizontal slice) does not always contain enough information to make an informed decision about whether the answer is certainly correct or incorrect. In fact, the heatmap shows that for any fixed verbal confidence value, the reasoning length can provide further information about whether the answer is correct or not. This complementarity leads to a useful way of combining length and verbal confidence, as we now discuss.

**A Simple Zero-Shot Combination Technique.** A simple way to incorporate the uncertainty information available in both verbalized confidence and the reasoning trace length is to simply take the normalized sum. Specifically, we first collect the entire set of verbalized confidences and trace lengths as features for each response in a dataset. Then, we scale these two features to have mean 0 and unit variance. Finally, we take the sum of the two features as the "uncertainty score". We note that this is zero-shot since it requires no gold-labels and can be obtained for free from the initial queries, and is also black-box as long as the reasoning trace is available.

In Figure 2, we show that this summing method, denoted VC+TL, performs better than using either VC or TL in isolation, when averaged over all prompts, datasets, and models. Complete results are available in Appendix F.1. In fact, for 32B models, VC+TL is the best performing method in 110 out of 120 cases across three prompts, four reasoning models, and ten datasets.

## 5 WHY IS LENGTH PREDICTIVE OF CONFIDENCE?

In this section we consider the natural followup question: why does length emerge as a useful confidence measure in reasoning models? We investigate three candidate explanations: (1) Relationship with the number of "forking tokens"; (2) Necessary changes in length due to problem difficulty; and (3) GRPO-specific training effects. We observe that trace length remains a strong signal even after controlling for problem difficulty and GRPO-induced length bias. Instead, we find that forking tokens are a key factor underlying its success.

### 5.1 THE ROLE OF FORKING TOKENS

Prior work has studied the important role that certain tokens such as "maybe" and "wait" play in reasoning (Yoon et al., 2025; Muennighoff et al., 2025; Vanhoyweghen et al., 2025) and non-reasoning (Bigelow et al., 2025) models. While such tokens have sometimes been termed "epistemic markers" (Liu et al., 2025a) or "linguistic hedges" (Tao et al., 2025) that intuitively indicate the LLM's uncertainty, this characterization risks anthropomorphization (Kambhampati et al., 2025). Rather, following Wang et al. (2025c), we focus on their role as "forking tokens" — tokens where the LLM often has high entropy in its token distribution.[2] These indicate "forks" in the generation process, as they are points where the output could have taken a very different route. In Appendix H, we show that common phrases like "maybe", "wait", and "perhaps" are indeed high-entropy forking tokens for the Skywork-OR1 reasoning model, and tokens like "sometimes" or "potentially" are forking tokens for Qwen2.5-Instruct.

Wang et al. (2025c) conduct a detailed analysis of forking tokens, observing that forking tokens form a small minority whose entropy reliably grows over the course of reasoning post-training. We postulate that amplifying the entropy of forking tokens is a natural way in which RL incentivizes exploration in a base model. This suggests that the usage of forking tokens can be seen as a direct expression of a model's operational uncertainty.

We argue that one key reason trace length is a useful uncertainty measure is that it is directly correlated with the number of such high entropy forking tokens. In the left of Figure 4, we demonstrate

---

[2]Formally, a token $t$ is a forking token if on average over all prefixes $p$ which precede $t$ in a dataset, the LLM's next-token distribution following $p$ has high entropy.

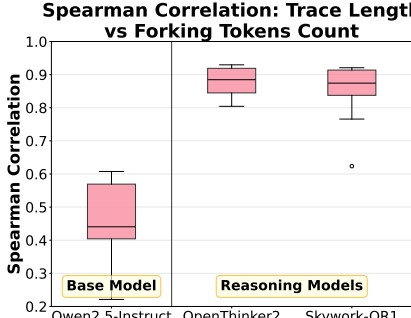

| OpenThinker2-32B | | | | | | |
|---|---|---|---|---|---|---|
| Dataset | TL | FT | TL+FT | SP | BFT | BestToken |
| Folktexts | 64.3 | **66.8** | 66.3 | 65.6 | 60.9 | considering |
| MediQ | 68.1 | 68.6 | 68.7 | **69.2** | 67.5 | maybe |
| MMLU-Pro NoMath | 78.5 | 79.9 | 79.6 | **80.1** | 79.9 | Alternatively |
| MedMCQA | 66.6 | 67.0 | 67.1 | **68.3** | 66.0 | maybe |
| MedQA | 79.5 | 78.9 | **80.3** | 80.1 | 77.1 | maybe |
| MMLU | 78.7 | 80.5 | 80.4 | **81.4** | 78.6 | maybe |
| SciQ | 74.9 | 75.7 | **76.3** | 75.6 | 74.3 | Alternatively |
| SuperGPQA | 61.6 | **62.8** | 62.5 | 62.5 | 61.9 | maybe |
| TriviaQA | 81.0 | 80.8 | 81.5 | **84.4** | 81.4 | maybe |
| NonambigQA | 74.6 | 76.6 | 76.4 | **77.0** | 76.3 | maybe |
| *Average* | 72.8 | 73.8 | 73.9 | **74.4** | 72.4 | - |

Figure 4: **Trace length strongly correlates with number of forking tokens in reasoning models.** (**Left**): For each 32B model, the box plot shows the distribution of spearman correlation values between trace length (in tokens) and the count of the top 50 highest entropy "forking" tokens for each dataset. Distribution is across ten datasets. Very high correlation is observed for the two reasoning models compared to the base model. (**Right**): Table showing the performance in AUROC of trace length (TL), top 50 highest entropy forking tokens (FT), the normalized sum TL+FT, and sequence probability (SP) for OpenThinker2-32B across ten datasets. We also include the AUROC of the best single forking token (BFT) over the dataset, and the BestToken itself. Generation details and additional tables are in Appendix I.

this strong correlation for two reasoning models; across ten datasets, the median Spearman correlation between the two values is well above 0.8. To dig into this relationship further, we conduct more detailed experiments:

1. **Performance of Forking Tokens vs. Trace Length.** On the right of Figure 4, we demonstrate that using the top 50 highest entropy tokens provide comparable AUROC to trace length for OpenThinker2-32B across ten datasets.[3] However, neither strictly dominates the other. In addition, Figure 4 demonstrates the existence of *single tokens* whose counts are competitive with the sequence probability in terms of AUROC. This is demonstrated by the best forking token (BFT) column, which selects the forking token whose count has the highest AUROC among the 50 highest entropy tokens. The token itself is also displayed.
2. **High Entropy Forking Tokens are Useful.** Broadly, we argue that the higher the entropy of a token, the more useful it is to include in the count. To demonstrate this, we add the highest-entropy tokens one by one to a "working set" of tokens to be counted, and observe that the AUROC of the count grows roughly monotonically until it matches trace length (depicted in Figure 5 for OpenThinker2-32B and Skywork-OR1-32B on two datasets). Additional plots are available in Appendix I.

**Truly Zero-Shot Methods.** We highlight that the methods discussed here, trace length and number of forking tokens, have a very significant advantage in practice: they do not even require changing the prompt. If we precompute the set of forking tokens offline, these methods and combinations thereof (such as TL+FT) can be applied to any black-box model *without intervening on the model at all*. Indeed, in Appendix G, we provide evidence that merely counting a small number of common forking tokens (or epistemic markers) performs comparably to trace length. From a scientific point of view, this sheds light on the ways LLMs express their uncertainty "in the wild", without any interventions. From a practical point of view, these methods are especially suitable for black-box use at inference time. This is in contrast to methods such as verbalized confidence, which requires (and is in fact sensitive to) prompt modifications, and sequence probability, which requires access to token probabilities.

## 5.2 THE (NON-)ROLE OF PROBLEM DIFFICULTY

A possible explanation for why length emerges as a useful uncertainty measure is that questions of differing difficulty may require a differing number of steps. To investigate this, we ask: controlling

---

[3]We define high entropy forking tokens per dataset, and only include tokens which appear in at least twenty unique responses in a dataset (see Appendix I for details) — this precludes tokens which are high entropy only because they are a possible answer to a particular question.

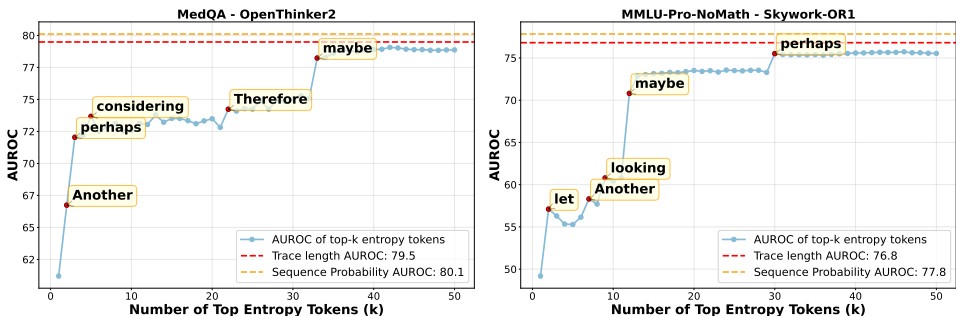

Figure 5: **High entropy tokens help quantify uncertainty in 32B models**. In each plot, we show the AUROC of the uncertainty score which counts the occurrence of any of the $k$ highest entropy forking tokens in each trace. As we increase $k$, the AUROC of the uncertainty score improves (see Section 5.1 for details). The AUROC of trace length and sequence probability are displayed for reference. Additional plots per model and dataset are available in Appendix I.

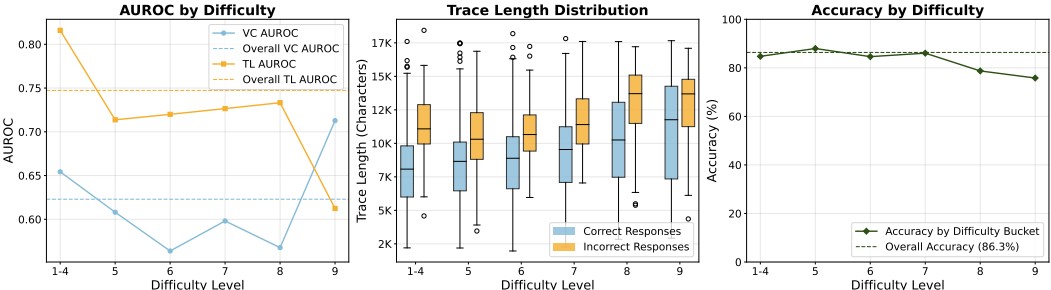

Figure 6: **The relationship between trace length and problem difficulty.** Evaluating OpenThinker2-32B on 10K questions from (He et al., 2025b) with Prompt 2, numeric confidence. (**Left**): AUROC of verbalized confidence (VC) and trace length (TL) by question difficulty level. Both TL and VC are still informative measures of correctness *within* each difficulty level, and also for the most difficult questions (level 9). (**Middle**): Mean and standard deviation of trace length for correct and incorrect responses by difficulty level. Correct trace lengths grow longer with more difficult questions.(**Right**): Accuracy by difficulty bucket.

for *similar difficulty*, does trace length still provide a useful uncertainty signal? If not, then trace length may just be providing a proxy for the intrinsic difficulty of the question.

We investigate this qualitatively in Figure 6, which uses the Deepmath-103k (He et al., 2025b) dataset. DeepMath classifies questions into different difficulty levels determined by carefully prompting and aggregating multiple GPT-4o responses. In the left of Figure 6, we demonstrate that trace length — and also verbalized confidence — provide better than random AUROC when looking at questions of similar difficulty. Furthermore, the spearman correlation between difficulty levels of DeepMath and TL is $\approx 0.3$, and between difficulty and VC $\approx 0.17$. These facts demonstrate that both TL and VC are capturing at least some information other than question difficulty.

In the middle plot of Figure 6, we also observe that the trace length of correct responses grows monotonically in the question difficulty. This provides additional justification for utilizing trace length as a *proxy* for question difficulty, as is often done in the SFT pipelines of models like OpenThinker (Guha et al., 2025). In such settings, the goal is often to collect traces for as difficult questions as possible. In particular, if you are collecting reasoning traces from R1 to distill a smaller model with, and you know that R1 is reasonably accurate on a particular dataset, trace length correlates well with question difficulty. In addition, since OpenThinker was trained with such a pipeline, this provides some explanation for why length is quite good for OpenThinker in particular, almost always out-performing verbalized confidence (see Appendix F.1).

## 5.3 THE (NON-)ROLE OF GRPO

In this section, we examine whether the correlation between length and correctness in post-trained models can be fully attributed to a length bias inherent in GRPO (Shao et al., 2024), as identified by

Liu et al. (2025c), who provide a comprehensive analysis of this bias. In brief, GRPO's objective function normalizes each response's advantage by its length, creating two opposing incentives. For correct responses with positive advantage, the model is encouraged to produce not only accurate but also concise answers. Conversely, for incorrect responses with negative advantage, the model is paradoxically incentivized to generate *longer* responses, as increasing the denominator reduces the magnitude of the negative contribution to the objective. These incentives could reasonably lead to what we see in practice: longer responses from the model are more likely to be incorrect.

We investigate the role of GRPO's length bias by comparing to a variant of GRPO, Dr. GRPO, proposed by Liu et al. (2025c). Dr. GRPO removes the length normalization term found in GRPO's objective in an attempt to reduce or eliminate this length bias.

We run both algorithms with the MATH dataset (Hendrycks et al., 2021b) on a base Qwen2.5-7B-Instruct model. Training and evaluation details are in Appendix D.1. The results, in Figure 7, demonstrate that reasoning trace length *still* emerges as a useful indicator for correctness, even with the length-bias corrections that Dr. GRPO provides. In particular, the right side of Figure 7 demonstrates that the mean correct and incorrect response length shift farther apart after running only 200 steps of Dr. GRPO on Qwen. In Appendix J, we show that a similar, even starker histogram is observed for Skywork-OR1-32B, which also removes the length normalization during RL post-training (similar to Dr. GRPO). Together, these results suggest that the underlying mechanism that makes length a useful proxy for correctness persists even when GRPO's length bias is removed.

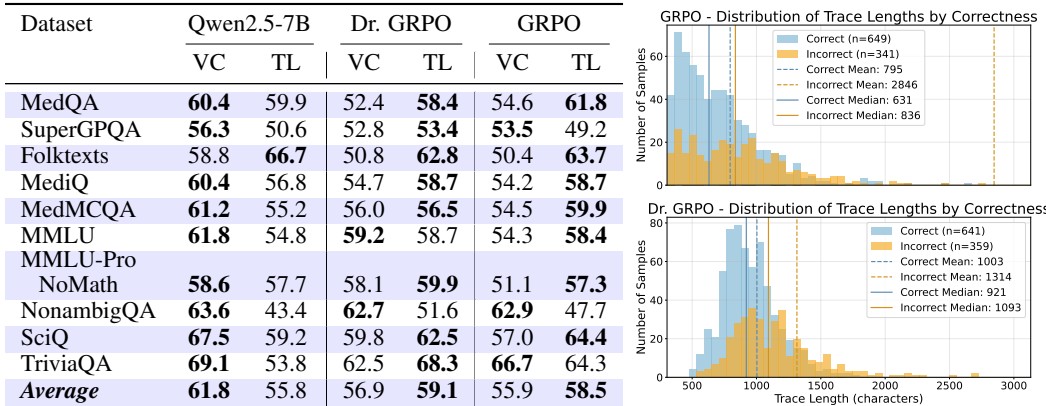

| Dataset | Qwen2.5-7B | | Dr. GRPO | | GRPO | |
|---|---|---|---|---|---|---|
| | VC | TL | VC | TL | VC | TL |
| MedQA | **60.4** | 59.9 | 52.4 | **58.4** | 54.6 | **61.8** |
| SuperGPQA | **56.3** | 50.6 | 52.8 | **53.4** | 53.5 | 49.2 |
| Folktexts | 58.8 | **66.7** | 50.8 | **62.8** | 50.4 | **63.7** |
| MediQ | **60.4** | 56.8 | 54.7 | **58.7** | 54.2 | **58.7** |
| MedMCQA | **61.2** | 55.2 | 56.0 | **56.5** | 54.5 | **59.9** |
| MMLU | **61.8** | 54.8 | **59.2** | 58.7 | 54.3 | **58.4** |
| MMLU-Pro NoMath | **58.6** | 57.7 | 58.1 | **59.9** | 51.1 | **57.3** |
| NonambigQA | **63.6** | 43.4 | **62.7** | 51.6 | **62.9** | 47.7 |
| SciQ | **67.5** | 59.2 | 59.8 | **62.5** | 57.0 | **64.4** |
| TriviaQA | **69.1** | 53.8 | 62.5 | **68.3** | **66.7** | 64.3 |
| *Average* | **61.8** | 55.8 | 56.9 | **59.1** | 55.9 | **58.5** |

Figure 7: **Trace length (TL) is still a useful quantity after Dr. GRPO. (Left)**: Impact of reinforcement learning using Dr. GRPO and GRPO on the effectiveness of TL in predicting correctness in a 7B model. AUROC of TL improves after RL on Qwen2.5-7B. (**Right**): Distribution of correct and incorrect answer on TriviaQA after RL with GRPO (above) and after Dr. GRPO (below). Mean of correct and incorrect answer lengths are still separated after Dr. GRPO, implying that TL is still a useful quantity for UQ.

## 6 LIMITATIONS AND CONCLUSION

Our results establish trace length as a simple and robust zero-shot confidence estimate for reasoning models that performs comparably to verbalized confidence while capturing complementary uncertainty information. Importantly, this signal is meaningful only after reasoning post-training, suggesting that post-training fundamentally alters how uncertainty manifests in model outputs. Our investigation into the mechanisms behind trace length reveals that its close connection to high-entropy "forking" tokens is a key driver of success. We believe the emergence of forking tokens as indicators of uncertainty represents a fundamental aspect of LLM UQ that warrants further investigation.

**Limitations.** First, while we test across multiple models, datasets, and prompts, the generalizability of our results to different model scales, base models, and training approaches remains to be established. Second, our investigation reveals that trace length may underperform as a confidence signal in extremely low-accuracy regimes. Our models and dataset range from 37% accuracy to 90% accuracy (Appendix F.4), but we observe that the AUROC of both trace length and verbal confidence tends to degrade on high-difficulty subsets of data where model accuracy is poor (Figure 6, Appendix K), consistent with findings from Vanhoyweghen et al. (2025). This suggests that alternative confidence signals may be more appropriate in domains where models struggle.

REPRODUCIBILITY STATEMENT

All the models we train and evaluate are detailed in Appendix C. All models selected for evaluation are open-weight, and most also share their training data. All datasets used in evaluation are listed in Appendix D, along with the particular HuggingFace splits used and any pre-processing steps taken. Each section of the appendix referenced by the main paper has the specific generation settings used (e.g., temperature, maximum token generation length, etc.). Our main evaluation code for verbal confidence is based on the available code of Yoon et al. (2025) (see Section 3). The training code for our GRPO and Dr. GRPO models is exactly the code made available by Liu et al. (2025c) with no modifications. The training code for iw-SFT (Qin & Springenberg, 2025) is exactly the code provided by the authors — we run it to generate our own version of the model.

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

## A  ADDITIONAL RELATED WORK DISCUSSION

In addition to the works discussed in Section 2 and throughout the paper, we discuss a few other relevant lines of work in this section.

**Effects of Reasoning Training on Confidence and Calibration**   First, while we focus on the contents of reasoning traces, we note two other works that broadly investigate changes in LLMs' verbalized confidence and calibration following reasoning training. Kirichenko et al. (2025) construct a benchmark dataset of questions where LLMs are sometimes expected to abstain or express low confidence when faced with unanswerable questions. After evaluating frontier reasoning models on this benchmark, they find that additional reasoning training appears to make LLMs overconfident and impairs their ability to appropriately abstain. Bereket & Leskovec (2025) also examine the calibration of LLMs trained with GRPO on tasks with stochastic outcomes, finding that GRPO similarly tends to make models overconfident. However, they demonstrate that alternative RL fine-tuning methods or removing certain biases from the GRPO objective can eliminate this overconfidence and yield better-calibrated models.

**Principled Evaluation Metrics for LLM Confidence**   Relevant to our discussion of evaluation metrics in Section 3.1, several other works have taken a critical approach to evaluating LLM expressions of confidence. Janiak et al. (2025) point out that while ROUGE scores are commonly used to grade LLM answers when assessing confidence, they are not always accurate and may misrepresent a model's true confidence quantification abilities. Following their recommendations, we use an LLM-as-Judge framework to evaluate the correctness of model responses. We note that Janiak et al. (2025) also consider response length as a signal for hallucinations in non-reasoning models. Taking a different perspective, Wang et al. (2025b) observe that different readers may interpret verbal expressions of uncertainty differently, arguing that rather than associating a single probability value with confidence expressions such as "very likely", we should associate a *distribution* of probability values to approximate the range of reader interpretations. They define and study new notions of calibration in this distributional setting. Finally, taking a broader perspective beyond language model confidence, Chidambaram & Ge (2024) discuss limitations of standard calibration metrics such as ECE when used as primary measures of calibration error.

**Fine-tuning for Improved Calibration**   Our work investigates confidence signals of LLMs that have not been explicitly fine-tuned for improved confidence estimates. However, we note a large body of work that focuses on fine-tuning for confidence estimation and overview a few key works here. Band et al. (2024) propose a method that incentivizes LLMs to include calibrated confidence estimates in long-form generations with many factual claims. Damani et al. (2025) and Li et al. (2025) both propose fine-tuning approaches based on optimizing a proper scoring rule, rather than a 0/1 correctness score, incentivizing the model to output calibrated confidence estimates. Lastly, Zhang et al. (2025b) consider fine-tuning for confidence in long form generations.

**Improving Performance and Efficiency with Confidence Estimates and/or Reasoning Trace Contents**   A number of works study how model accuracy or efficiency can be improved by leveraging confidence estimates or reasoning trace contents. Taubenfeld et al. (2025) leverage confidence estimates to improve inference-time self-consistency techniques. Fu et al. (2025) similarly use confidence estimates to propose a method that filters out low-quality generations at test time. Focusing on improving efficiency as measured by reasoning trace length, Wang et al. (2025a) explore ways to shorten reasoning traces without degrading model accuracy. Zhu & Li (2025) present a survey of similar works that attempt to shorten the traces of reasoning models.

## B  FURTHER DISCUSSION OF METRICS EVALUATING LLM UNCERTAINTY

This section presents a more detailed discussion of the points overviewed in Section 3.1.

### B.1  FORMAL DEFINITIONS

We first formally define each measure. We describe how to evaluate each metric with respect to a test set $D_{test} = \{(x_i, y_i, p_i)\}_{i=1}^n$ where $x_i$ is a prompted question and model-generated answer, $y_i \in$

$\{0, 1\}$ is a binary label denoting whether the model's answer was correct, and $p_i \in \{0, 0.1, ..., 0.9, 1\}$ is the model-generated confidence in its answer. With this setup in mind, we can define each of the four metrics.

**Definition B.1** (Accuracy). The model's accuracy on the test set, $\mathsf{Acc}(D_{test})$, is exactly the proportion of questions that it answered correctly.

$$\mathsf{Acc}(D_{test}) := \frac{1}{n} \sum_{i=1}^{n} y_i.$$

**Definition B.2** (Brier Score). The Brier Score of the model on the test set is the squared error of the model's confidence values in predicting the correctness of the answer:

$$\mathsf{Brier}(D_{test}) = \frac{1}{n} \sum_{i=1}^{n} (y_i - p_i)^2.$$

**Definition B.3** (Expected Calibration Error (ECE)). The ECE measures the expected difference between accuracy and confidence across the model's confidence bins.

$$\mathsf{ECE}(D_{test}) = \sum_{p \in \{0, 0.1, ..., 0.9, 1\}} \left| \frac{1}{n} \sum_{i=1}^{n} \mathbf{1}[p_i = p](p_i - y_i) \right|.$$

**Definition B.4** (The Receiving Operating Characteristic (ROC) Curve and Area Under the Curve (AUROC)). The ROC curve and associated AUROC measure how well the model's confidence values allow for distinguishing correct and incorrect answers.

The ROC curve of a model on a test set is obtained by considering various thresholds $\tau \in [0, 1]$, and plotting the resulting true positive rate (TPR) and false positive rate (FPR) along $\tau$, i.e. the parametrized curve $\{(\mathsf{FPR}(p, \tau), \mathsf{TPR}(p, \tau)\}_{\tau \in [0,1]]}$ where

$$\mathsf{FPR}(p, \tau) = \frac{1}{\sum_{i=1}^{n}(1 - y_i)} \sum_{i=1}^{n} \mathbf{1}[p_i > \tau](1 - y_i), \quad \mathsf{TPR}(p, \tau) = \frac{1}{\sum_{i=1}^{n} y_i} \sum_{i=1}^{n} \mathbf{1}[p_i > \tau] y_i.$$

The AUROC of $p$ is defined as the area between the line $y = 0$ and the ROC curve between $x = 0$ and $x = 1$. A perfect classifier will have an AUROC of 1, whereas a completely random classifier has an AUROC of 1/2.

We point out that AUROC can still be computed when the scores $p_i$ lie in $\mathbb{R}$, and are not necessarily probabilities. In this case, we vary over all thresholds $\tau \in \mathbb{R}$ which divide the samples into two sets of positive or negative predicted class. The remainder of the computation is identical.

### B.2 EXTENDED DISCUSSION

Below we elaborate on the arguments outlined in 3.1. See also (Chidambaram & Ge, 2024) for related discussion of the pitfalls of using ECE and variants as measures of calibration error.

**ECE and AUROC are not necessarily aligned**   Consider the common scenario depicted in Figure 8: Model A exhibits lower ECE than Model B, suggesting better calibration, yet Model A also shows lower AUROC, indicating worse discrimination ability. This simple example illustrates a more fundamental problem: each of these metrics capture different aspects of UQ and accuracy, making it unclear which should take precedence when they disagree.

**Uncertainty Estimates Should Help *Distinguish* Between High and Low Uncertainty**   The fundamental purpose of uncertainty quantification is to identify when we should trust a model's outputs versus when we should remain skeptical. Effective uncertainty estimates must therefore *distinguish* between correct and incorrect predictions—a capability that different metrics reward to varying degrees.

Both Brier score and AUROC directly reward uncertainty estimates that effectively discriminate between correct and incorrect predictions. In contrast, ECE is *not* tied to the discriminative ability of

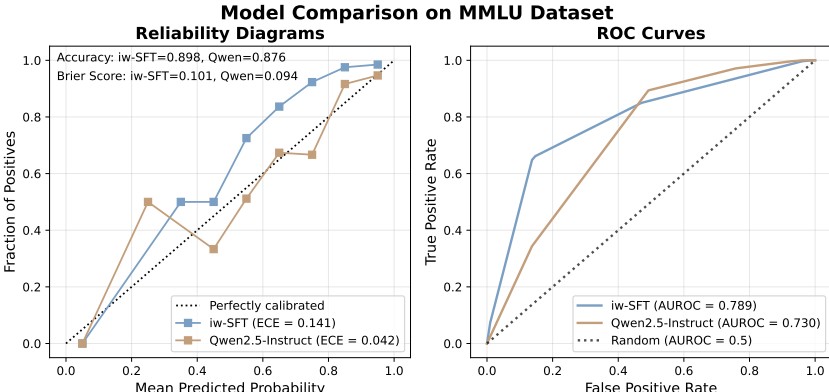

Figure 8: **ECE and AUROC can provide conflicting assessments of confidence estimate quality.** Using Prompt 1, we compare the verbalized confidence abilities on MMLU (Wang et al., 2024b) of Qwen2.5-32B-Instruct and iw-SFT (Qin & Springenberg, 2025), a reasoning model SFT'd from Qwen2.5-32B-Instruct. Models have similar accuracy and Brier score, however, iw-SFT has better AUROC, and Qwen better ECE. Which model has better uncertainty estimates?

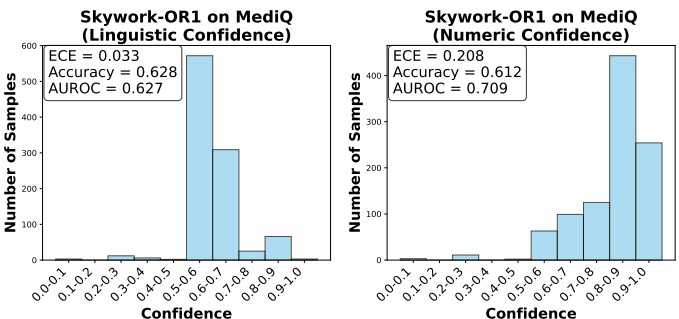

Figure 9: **ECE can inappropriately reward uninformative confidence estimates that fail to distinguish between correct and incorrect answers.** We compare verbal confidence estimates using Prompt 1 (**Left**) and numeric estimates using Prompt 2 (**Right**) for Skywork-OR1 on MediQ. The verbal estimates achieve extremely low ECE but only because the confidence distribution is highly peaked around the model's overall accuracy, providing minimal discriminative information. In contrast, the numeric prompt yields higher ECE due to a more diverse confidence distribution, but this diversity enables substantially better discrimination between correct and incorrect responses, as evidenced by the higher AUROC. This illustrates how ECE can misleadingly favor concentrated confidence distributions that happen to align with dataset accuracy, while penalizing genuinely useful uncertainty estimates with superior discriminative ability.

uncertainty estimates. ECE measures only whether confidence levels align with empirical accuracy within predefined bins, ignoring whether the model can meaningfully differentiate between high and low confidence cases. This can lead ECE to paradoxically reward uninformative estimates while penalizing genuinely useful ones.

Consider the following illustrative example: on a dataset where a model achieves 70% accuracy, an uncertainty estimate that uniformly assigns 0.7 confidence to *all* responses, regardless of correctness, receives a perfect ECE of 0. Meanwhile, an estimate that assigns 0.5 confidence to incorrect answers and 0.6 to correct answers achieves a (poor) ECE of 0.43, despite providing substantially more actionable information that can be used to discriminate between responses of different qualities. This pathology appears in practice: in Figure 9, we present a model that achieves an ECE of 0.15 on TriviaQA but 0.33 on NonambigQA, with this difference driven purely by overall dataset accuracy rather than improved uncertainty quantification.

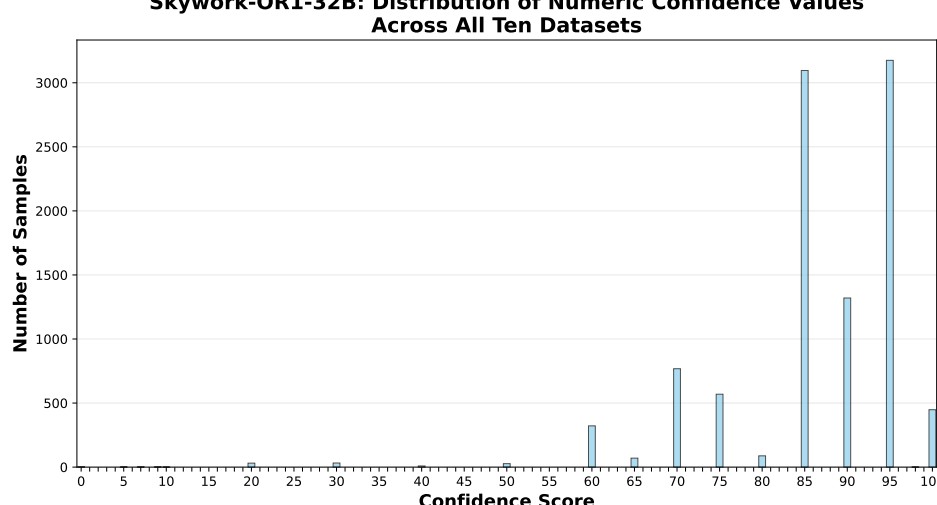

Figure 10: **Reasoning models exhibit numeric bias in their verbalized confidence estimates.** Histogram showing the outputted confidence values for Skywork-OR1 over all 10 datasets when prompted with Prompt 2. While Prompt 2 allows the model to output any integer value from 0-100, Skywork-OR1 exclusively outputs multiples of 5, with high concentration on values above 60, particularly 85 and 95.

**AUROC Avoids the Pitfalls of Value-Based Metrics**  While both Brier score and AUROC reward discriminative ability, Brier score suffers from being a *value-based* metric that requires precise probability calibration. This creates fundamental problems for evaluating confidence estimates that don't naturally output precise probabilities, such as verbalized confidence approaches where models are asked asks to express uncertainty through linguistic phrases like "highly likely" or "unlikely" (see Prompt 1 or Wang et al. (2025b); Yoon et al. (2025); Yang et al. (2024b)) or implicit confidence signals derived from response characteristics such as trace length.

Value-based metrics introduce two critical issues for non-probabilistic confidence estimates. First, the mapping from linguistic phrases to probability values is inherently arbitrary—should "Almost Certain" correspond to 0.95, 0.90, or 0.99? Different mappings can dramatically alter metric scores, meaning value-based metrics may reward fortuitous mapping choices rather than genuine uncertainty quantification ability. This concern is validated by Yoon et al. (2025), who perform an ablation removing the probability values from their verbalized confidence prompt (Prompt 1). Despite the model no longer seeing explicit probability-phrase mappings, confidence estimate performance remains largely unchanged (Table 7 of Yoon et al. (2025)), suggesting that the original prompt's success stemmed from fortuitous alignment between the chosen probability assignments and the model's natural interpretation of uncertainty phrases, rather than the model actually following the instructed numerical values.

Second, even when LLMs do output numerical probabilities directly, they often exhibit systematic biases towards numbers more likely to appear in a pre-training corpus. For example, Stureborg et al. (2024) show that when asking GPT-3.5 to grade textual summaries on a scale from 1-100, the model output 95 more than 20% of the time, whereas odd numbers like 92 or 89 are hardly ever output. In addition, the model never outputs any score from 1-60. When asking for verbal confidence as a number, a similar bias appears, as illustrated in 10. Non-probabilistic confidence signals may allow LLMs to communicate uncertainty more naturally without these numerical artifacts, but value-based metrics cannot capture this potential advantage since they require converting all expressions back to numerical probabilities.

AUROC sidesteps these issues entirely by evaluating only the relative ordering of confidence estimates, making it robust to arbitrary probability mappings and applicable even to confidence signals such as trace length that resist natural conversion to probability values.

*Remark* B.1. A more principled approach to probability assignment to confidence signals would be isotonic regression on a held-out calibration dataset. However, it's important to note that this will drive ECE to zero in all cases, while leaving AUROC largely unchanged. This further demonstrates the way in which ECE conflates discriminative ability with mapping choices, rewarding arbitrary decisions that happen to align with dataset statistics rather than genuine uncertainty quantification ability.

*Remark* B.2. A potential limitation of AUROC is that it may not compose well across heterogeneous datasets when confidence scales differ systematically between tasks. In particular, if one question type naturally elicits longer chain-of-thought responses than another (independent of actual uncertainty), two datasets could each exhibit strong AUROC for trace length individually, yet show poor AUROC on their union if one dataset's scale dominates the other. This is in contrast to value-based metrics such as ECE or Brier score, which average across combined datasets.

Due to this concern, when deploying uncertainty-based decision systems, we recommend stratifying by task type where possible to account for potential scale confounds unrelated to genuine uncertainty and extract the greatest possible discriminative ability from uncertainty metrics such as trace length.

## C   MODEL DETAILS

In this section, we detail each of the models we evaluate. For all models, unless specified otherwise for the particular experiment or model, we evaluate with temperature 0.

**OpenThinker2-32B** (Guha et al., 2025): This model is trained by SFTing Qwen2.5-32B-Instruct on more than 100k reasoning carefully selected traces generated by the full Deepseek-R1 model (Guo et al., 2025).

**iw-SFT-32B** (Qin & Springenberg, 2025): This model was trained by SFTing Qwen2.5-32B-Instruct on the s1.1 dataset (Muennighoff et al., 2025). Ths s1.1 dataset contains around 1k difficult questions with reasoning traces generated by Deepseek-R1. Importantly, the model is trained using an importance weighted (iw) variant of cross-entropy loss, which they claim for particular weights makes SFT closer to RL post-training. The authors have posted a version of the model on huggingface. Instead of using this version, we run the code provided by the authors in order to reproduce the model.

As a side effect of using the importance weighted cross-entropy loss for SFT, the iw-SFT model does not require "budget forcing" by adding "wait" tokens to force additional reasoning (Qin & Springenberg, 2025). Such an approach was necessary in order to elicit optimal performance when using standard SFT on the original s1.1 dataset (Muennighoff et al., 2025).

**R1-Distill-32B** (Guo et al., 2025): This is a version of Qwen2.5-32B (base model, non-instruct) which is SFT'd on traces generated by R1.

**Skywork-OR1-32B** (He et al., 2025a): This is a version of R1-Distill-32B with GRPO applied directly on top. The data and training code are available in the technical report. Interestingly, the model is post-trained with a version of GRPO which corrects for the length bias (Liu et al., 2025c).

**OpenReasoning-Nemotron-7B** (Nathawani et al., 2025): This is a version of Qwen2.5-7B (base model, non-instruct) SFT'd on R1 traces.

**OpenThinker3-7B** (Guha et al., 2025): This model is a version of Qwen2.5-7B-Instruct SFT'd on 1.2 million carefully selected R1 reasoning traces. We found that setting temperature to 0 for this model provided poor performance (getting stuck in reasoning loops). Instead, throughout all experiments for this model, we used the recommended generation parameters provided in the huggingface repository. Namely, temperature 0.7, repetition penalty 1.05, top-k of 20, and top-p of 0.8.

**LLM as a Judge**: We use Qwen2.5-32B-Instruct as a judge throughout all our experiments. We use the same judge prompt and judging framework as (Yoon et al., 2025). In particular, we feed the set of possible answers, and a standard comprehensive judge prompt to Qwen. While (Yoon et al., 2025) used gpt-4o as a judge, we found similar results when using Qwen2.5-32B-Instruct.

# D   DATASET DETAILS

In this section, we provide citations and curation details for all datasets evaluated on. For each dataset (except DeepMath-130k), we select a random subset of 1000 examples to evaluate on. We detail the exact split on huggingface used for each dataset.

**NonAmbigQA** (Yoon et al., 2025): Subset of AmbigQA dataset (Min et al., 2020) created by (Yoon et al., 2025). The subset is created by selecting 1000 non-ambiguous questions from AmbigQA, which itself was created by looking at Natural Questions (Kwiatkowski et al., 2019).

**MMLU** (Hendrycks et al., 2021a;b): Classic multiple choice response dataset. We use the "all" subset, and the validation split for evaluation.

**MediQ** (Li et al., 2024): A set of contextual situations with a required patient diagnosis at the end. The model is given the prompt: "'The following is a list of medical facts, some of which pertain to a patient", followed by the first three atomic facts for each question in the dataset. Then, the question asked is "Using these facts, answer the following multiple choice question:", followed by the question provided in the dataset. We use the validation split for evaluation.

**MedMCQA** (Pal et al., 2022): Large scale multiple choice questions from real-world clinical exams. We filter to looking at a random subset of questions with only a single correct answer from the validation set.

**SciQ** (Johannes Welbl, 2017): Simple multiple choice science questions (4 choices per question). We use randomly selected examples from the default split on huggingface.

**MedQA** (Jin et al., 2020): Multiple choice questions with 4 options focused on identifying the best treatment plan given the background and details of a patient. We use randomly selected examples from the train split on huggingface.

**MMLU-Pro NoMath** (Wang et al., 2024a): Multiple choice questions with 10 options across disciplines. This is a subset of MMLU-Pro which only includes problems *without* multi-step reasoning (as determined by Claude). We randomly select all examples from the test split on huggingface.

**TriviaQA** (Joshi et al., 2017): Short answer trivia questions. We randomly select all examples from the "rc" subset of the validation split from huggingface.

**SuperGPQA** (Team et al., 2025a): Challenging multiple choice questions across many different disciplines. We randomly select all examples from the train split on huggingface.

As suggested by Devic et al. (2025), we also include a dataset with natural intrinsic aleatoric uncertainty.

**FolkTexts** (Cruz et al., 2024): Folktexts is a dataset of questions based on US Census data. We use the "ACSIncome" task and the validation split from huggingface. The dataset is multiple choice, with two choices per question. Each question provides a number of details about an individual, including their age, education, sex, etc. The question then asks to predict whether the income of the individual is below or above a threshold. There is a natural amount of aleatoric uncertainty involved, since two individuals with similar features may have different labels. Thus, models should potentially output lower confidence to reflect this.

**DeepMath-103k**: In Section 5.2, we utilize the DeepMath dataset since it has difficulty annotations for each problem (generated by Claude). We use 10k examples randomly selected from the train split. The response format required by DeepMath is a single number or equation for each problem.

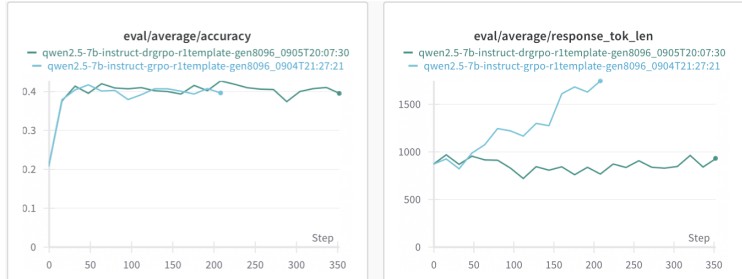

Figure 11: **Dr. GRPO effectively addresses the response length bias of GRPO.** Screenshot of wandb runs for GRPO and Dr. GRPO. The left shows evaluation performance (we select step 192 for the evaluation of both models). The right side demonstrates that Dr. GRPO effectively curtails the length bias of GRPO.

## D.1  TRAINING AND EVALUATING THE GRPO AND DR. GRPO MODELS

In Section 5.3, we demonstrate the performance of two post-trained models relative to a Qwen baseline model, one using GRPO and the other using Dr. GRPO. Both models are versions of Qwen2.5-7B-Instruct trained using GRPO or Dr. GRPO. Both models were also generated with the original code of the original Dr. GRPO authors (Liu et al., 2025c), with no modifications except maximum generation length. We run both algorithms for at least 200 steps and select step 192 for evaluation, as the eval accuracy is similar for both models there.

We train and evaluate both models with a maximum generation length of 8192. All other parameters are kept as the original parameters from the authors on the github repository. The training data used for both models is levels 3 through 5 of the MATH dataset (Hendrycks et al., 2021b). We use a simple 0-1 reward for correctness. The eval set contains challenging mathematical reasoning benchmarks such as AIME.

In Figure 11, we demonstrate that the average eval set response length in tokens of Dr. GRPO is indeed much lower than that of GRPO, demonstrating that Dr. GRPO effectively addresses the response length bias. We note that we only noticed this with a large enough maximum generation length during training — too small a length and both models had similar average response lengths near the maximum allowed generation length.

Evaluations in Figure 7 were done with temperature 0 and response length 8192.

# E    PROMPTS USED

In this section, we list out the standard prompt and verbalized confidence prompts used to run our evaluations. We begin with three verbalized confidence prompts. The first prompt, Prompt 1, is taken from (Yoon et al., 2025). The intuition is that models may be better at expressing their confidence in "words", or linguistic confidence phrases, than with numbers.

---

**Prompt 1: Verbal Confidence Prompt (Linguistic Uncertainty)**

Begin with a <think> tag and reason through the question step by step to arrive at an answer.
Then, thoroughly assess your confidence in that answer by evaluating your thinking process so far.
End your thinking process with a </think> tag.
Finally, classify your confidence into one of the following classes based on how likely your answer is to be correct:

- "Almost no chance" (0.0–0.1)
- "Highly unlikely" (0.1–0.2)
- "Chances are slight" (0.2–0.3)
- "Unlikely" (0.3–0.4)
- "Less than even" (0.4–0.5)
- "Better than even" (0.5–0.6)
- "Likely" (0.6–0.7)
- "Very good chance" (0.7–0.8)
- "Highly likely" (0.8–0.9)
- "Almost certain" (0.9–1.0)

Each category reflects the probability that your answer is correct.

At the very end of your output, format your answer and confidence as
**Answer**: $ANSWER
**Confidence**: $CLASS
where CLASS is one of the names (only the names without the probability ranges) of the classes above, and ANSWER is your final answer stated as concisely as possible.

---

When using Prompt 1 (or any of the following prompts) with a base, non-reasoning model (e.g., Qwen2.5-Instruct), we exclude instructions to generate the <think> tags. When we measure calibration error of models using Prompt 1, we map the linguistic confidence $CLASS output by the model to the *middle* of the corresponding bucket given to the model in the prompt. For example, if the model output "very good chance", we say that the model output probability 0.75.

The second prompt, Prompt 2, is a variation of the standard / chain-of-thought numeric confidence elicitation prompt from Mei et al. (2025).

---

**Prompt 2: Verbal Confidence Prompt (Numeric Uncertainty)**

Begin with a <think> tag and reason through the question step by step to arrive at an answer.
Then, thoroughly assess your confidence in that answer by evaluating your thinking process so far.
End your thinking process with a </think> tag.
Finally, return your confidence as an integer between 0 and 100 based on how likely your answer is to be correct.
That is, if your confidence is 0, that means that your answer has almost no chance of being correct.
If your confidence is 100, then you are almost certain that your answer is correct.

At the very end of your output, format your answer and confidence as:
**Answer**: $ANSWER
**Confidence**: $CONFIDENCE
where CONFIDENCE is an integer between 0 and 100, and ANSWER is your final answer stated as concisely as possible.

---

The final prompt we test is the top-$k$ prompt adapted from Mei et al. (2025); Tian et al. (2023).

---

**Prompt 3: Verbal Confidence Prompt (Top-K)**

Begin with a <think> tag. Give your K = 5 best guesses to the following question, and also your confidence in each guess (i.e., the probability that each one is correct).
If there are less than 5 possible answers, simply go through all possible answers and give your confidence in each.
Once you have given your K = 5 best guesses and their confidences, end with a </think> tag.
Finally, give your final answer and confidence in the following format:
**Answer**: $ANSWER
**Confidence**: $CONFIDENCE
where CONFIDENCE is an integer between 0 and 100, and ANSWER is your final answer stated as concisely as possible.

---

Next, we present the answer-only prompt used in Appendix F.3. Again, we remove the think tags if we run it with a non-reasoning model.

---

**Prompt 4: Answer-only Prompt**

Begin with a <think> tag and reason through the question step by step to arrive at an answer. At the very end of your output, format your answer as:
**Answer**: $ANSWER
where ANSWER is your final answer stated as concisely as possible.

---

# F    FULL RESULTS

Throughout this section, we evaluate all models with temperature 0. The exception is OpenThinker3-7B which, as we stated in Appendix C, we evaluate with temperature 0.7 and default generation arguments (we found that it was not succeeding at temperature 0). We use generation length 4096 for both the 32B models and the 7B models.

## F.1    RESULTS FOR 32B PARAMETER MODELS

| Dataset | Qwen2.5-Instruct | | | iw-SFT | | | OpenThinker2 | | | Skywork-OR1 | | | R1-Distill | | |
|---|---|---|---|---|---|---|---|---|---|---|---|---|---|---|---|
| | VC | TL | VC+TL | VC | TL | VC+TL | VC | TL | VC+TL | VC | TL | VC+TL | VC | TL | VC+TL |
| MedQA | **67.6** | 58.9 | 67.3 | 71.7 | 81.0 | **81.3** | 69.6 | 80.0 | **80.3** | 71.7 | **81.4** | 81.3 | 63.8 | 73.8 | **75.7** |
| SuperGPQA | **61.1** | 51.6 | 60.1 | 61.0 | 58.4 | **61.4** | 57.3 | 59.6 | **61.1** | 56.5 | 57.2 | **58.9** | 55.9 | 57.3 | **60.4** |
| Folktexts | 51.9 | 60.1 | **60.9** | 64.4 | 72.2 | **74.7** | 63.1 | 63.9 | **67.7** | 53.5 | 58.0 | **59.8** | 62.0 | 59.0 | **65.0** |
| MediQ | **66.8** | 56.9 | 65.7 | 63.0 | **71.4** | 71.3 | 65.1 | 69.7 | **70.5** | 62.7 | 66.7 | **68.2** | 60.1 | 65.7 | **67.4** |
| MedMCQA | **70.7** | 57.6 | 69.2 | 69.3 | 76.5 | **77.0** | 66.5 | 71.3 | **72.4** | 65.6 | 67.3 | **68.8** | 66.1 | 67.0 | **71.8** |
| MMLU | 73.0 | 59.4 | **74.2** | 78.9 | 82.9 | **84.3** | 76.4 | 82.5 | **84.7** | 71.8 | 76.9 | **79.7** | 67.5 | 70.0 | **75.2** |
| MMLU-Pro NoMath | **69.8** | 57.3 | 68.3 | 71.2 | 79.0 | **79.2** | 70.6 | 79.1 | **80.2** | 67.9 | 76.8 | **77.2** | 62.9 | **76.2** | 75.9 |
| NonambigQA | 72.1 | 51.3 | 63.3 | 73.3 | 74.5 | **77.2** | 73.1 | 73.9 | **77.4** | 72.6 | 72.6 | **76.9** | 74.2 | 67.8 | **76.0** |
| SciQ | 68.9 | 58.7 | 67.4 | 75.2 | 76.0 | **78.4** | 71.2 | 77.9 | **79.2** | 72.6 | 72.3 | **75.5** | 69.7 | 66.7 | **71.8** |
| TriviaQA | 77.4 | 60.8 | 76.7 | 82.2 | 84.5 | **86.4** | 78.1 | 85.1 | **86.9** | 81.0 | 80.3 | **85.5** | 78.5 | 75.1 | **81.8** |
| *Average* | 67.9 | 57.3 | 67.3 | 71.0 | 75.6 | **77.1** | 69.1 | 74.3 | **76.0** | 67.6 | 70.9 | **73.2** | 66.1 | 67.9 | **72.1** |

Table 1: AUROC comparison for VC, TL, VC+TL metrics for 32B models using the *linguistic confidence* prompt (i.e., asking the model to output the phrase which best expresses its confidence in the answer, see Prompt 1 in Appendix E). AUROC values are multiplied by 100 for readability. Best value for each model on each dataset is bolded.

| Dataset | Qwen2.5-Instruct | | | iw-SFT | | | OpenThinker2 | | | Skywork-OR1 | | | R1-Distill | | |
|---|---|---|---|---|---|---|---|---|---|---|---|---|---|---|---|
| | VC | TL | VC+TL | VC | TL | VC+TL | VC | TL | VC+TL | VC | TL | VC+TL | VC | TL | VC+TL |
| MedQA | **71.4** | 60.7 | 68.4 | 78.5 | 80.1 | **82.2** | 73.9 | 78.4 | **80.3** | 82.4 | 80.9 | **84.0** | 77.2 | 75.9 | **79.2** |
| SuperGPQA | 57.3 | 51.7 | 55.2 | 59.8 | 57.8 | **60.7** | 62.2 | 58.4 | **63.0** | 60.6 | 57.1 | **61.3** | 62.7 | 60.1 | **66.5** |
| Folktexts | 68.1 | 62.6 | **69.3** | 72.4 | 69.3 | **74.0** | 68.4 | 64.1 | **69.3** | **72.8** | 56.4 | 69.1 | 72.8 | 62.5 | **73.0** |
| MediQ | 66.9 | 58.7 | 65.9 | 70.6 | 70.1 | **71.9** | 65.8 | 70.2 | **71.2** | 70.9 | 70.5 | **72.5** | 67.7 | 67.0 | **69.6** |
| MedMCQA | 69.9 | 62.4 | 69.4 | 71.1 | 72.0 | **73.8** | 71.7 | 71.5 | **74.1** | 72.0 | 71.5 | **74.2** | 71.1 | 64.5 | **71.6** |
| MMLU | 72.0 | 65.3 | **73.0** | 78.4 | 82.2 | **83.1** | 79.2 | 81.6 | **83.8** | 83.2 | 78.8 | **85.0** | 76.1 | 72.2 | **78.9** |
| MMLU-Pro NoMath | **71.1** | 60.1 | 69.6 | 75.9 | 77.4 | **79.5** | 73.2 | 80.2 | **80.8** | 73.9 | 77.6 | **80.1** | 69.3 | 76.4 | **78.6** |
| NonambigQA | 66.7 | 53.1 | 61.9 | **79.0** | 73.7 | 78.8 | 77.9 | 73.2 | **78.6** | **80.0** | 72.9 | 79.7 | **75.7** | 66.8 | 75.4 |
| SciQ | 67.3 | 62.8 | **67.4** | 77.7 | 74.5 | **78.5** | 73.3 | 75.8 | **78.2** | 79.8 | 76.0 | **80.4** | 71.8 | 66.7 | **72.2** |
| TriviaQA | 72.8 | 60.8 | **73.5** | 81.4 | 81.3 | **83.9** | 86.2 | 84.1 | **88.2** | 84.1 | 79.1 | **84.3** | 79.3 | 72.6 | **80.7** |
| *Average* | 68.4 | 59.8 | 67.4 | 74.5 | 73.8 | **76.6** | 73.2 | 73.8 | **76.8** | 76.0 | 72.1 | **77.1** | 72.4 | 68.5 | **74.6** |

Table 2: AUROC comparison for VC, TL, VC+TL metrics for 32B models using the *numeric* confidence prompt (i.e., asking the model to output a confidence score in [0,100], see Prompt 2 in Appendix E). AUROC values are multiplied by 100 for readability. Best value for each model on each dataset is bolded.

| Dataset | Qwen2.5-Instruct | | | iw-SFT | | | OpenThinker2 | | | Skywork-OR1 | | | R1-Distill | | |
|---|---|---|---|---|---|---|---|---|---|---|---|---|---|---|---|
| | VC | TL | VC+TL | VC | TL | VC+TL | VC | TL | VC+TL | VC | TL | VC+TL | VC | TL | VC+TL |
| MedQA | **71.7** | 59.9 | 70.1 | 69.7 | 76.0 | **76.3** | 77.8 | 81.0 | **82.6** | 77.8 | 82.3 | **83.4** | 74.3 | 73.4 | **77.7** |
| SuperGPQA | **56.7** | 51.5 | 56.1 | 61.8 | 56.3 | **62.2** | 61.4 | 63.8 | **65.6** | 59.6 | 60.0 | **64.1** | 59.4 | 55.4 | **61.9** |
| Folktexts | **68.7** | 57.9 | 67.3 | 70.5 | 59.3 | 68.7 | 73.8 | 67.8 | **74.3** | 72.8 | 61.0 | 71.7 | **74.2** | 59.1 | 72.2 |
| MediQ | 63.2 | 56.8 | 62.5 | 64.9 | 70.4 | **71.1** | 68.4 | 72.2 | **73.2** | 69.7 | 68.9 | **71.7** | 64.7 | 61.5 | **66.0** |
| MedMCQA | 70.7 | 64.1 | **72.3** | 71.8 | 71.6 | **75.2** | 73.1 | 74.4 | **77.1** | 71.4 | 69.1 | **73.7** | 67.6 | 60.7 | **68.2** |
| MMLU | 73.7 | 69.3 | **79.0** | 77.7 | 77.0 | **81.3** | 81.0 | 80.7 | **85.1** | 78.6 | 75.5 | **81.9** | 75.0 | 66.8 | **77.3** |
| MMLU-Pro NoMath | 69.9 | 63.9 | **72.4** | 68.5 | 72.7 | **75.2** | 72.7 | 77.4 | **80.1** | 73.1 | 76.3 | **79.6** | 69.3 | 68.6 | **75.0** |
| NonambigQA | 68.2 | 67.1 | **71.2** | 76.9 | 70.1 | **77.8** | 77.1 | 73.2 | **78.6** | 80.1 | 69.1 | 78.4 | 75.4 | 65.9 | **75.5** |
| SciQ | **74.9** | 64.2 | 72.3 | 73.8 | 72.5 | **77.4** | 77.4 | 82.1 | **84.4** | 73.2 | 71.6 | **75.3** | 68.9 | 67.0 | **72.1** |
| TriviaQA | 75.8 | 75.9 | **81.5** | 84.2 | 77.6 | **85.3** | 84.2 | 81.5 | **86.7** | 85.2 | 79.9 | **86.2** | 79.7 | 72.3 | **81.3** |
| *Average* | 69.3 | 63.1 | **70.5** | 72.0 | 70.3 | **75.0** | 74.7 | 75.4 | **78.8** | 74.1 | 71.4 | **76.6** | 70.9 | 65.1 | **72.7** |

Table 3: AUROC comparison for VC, TL, VC+TL metrics for 32B models using the *top-k* prompt (i.e., asking the model to output the top-k most likely answers then choosing the one with the highest confidence score; see Prompt 3 in Appendix E). AUROC values are multiplied by 100 for readability. Best value for each model on each dataset is bolded.

## F.2 RESULTS FOR 7B PARAMETER MODELS

In this section, we present results for the three prompt in Appendix E on two 7B parameter models derived from Qwen2.5-7B-Instruct.

| Dataset | Qwen2.5-7B-Instruct | | | Nemotron-7B | | | OpenThinker3-7B | | |
|---|---|---|---|---|---|---|---|---|---|
| | VC | TL | VC+TL | VC | TL | VC+TL | VC | TL | VC+TL |
| MedQA | 61.0 | 60.8 | **64.6** | 71.1 | 75.0 | **76.5** | 54.4 | 59.5 | **59.8** |
| SuperGPQA | **56.2** | 50.9 | 54.8 | **58.8** | 48.9 | 54.2 | 50.1 | **53.3** | 52.7 |
| Folktexts | 57.8 | 64.2 | **64.5** | 63.8 | 64.1 | **65.6** | 37.0 | **60.0** | 45.7 |
| MediQ | 60.2 | 58.3 | **62.1** | 69.2 | 69.0 | **71.5** | 54.6 | **58.4** | 57.8 |
| MedMCQA | 61.3 | 56.1 | **61.8** | 63.5 | 65.8 | **67.3** | 57.5 | 63.3 | **64.1** |
| MMLU | **64.0** | 56.3 | 63.0 | 68.2 | 74.1 | **77.4** | 64.2 | 68.8 | **73.4** |
| MMLU-Pro | | | | | | | | | |
| NoMath | 58.0 | 57.7 | **60.5** | 66.5 | 73.8 | **75.5** | 56.4 | 65.3 | **65.8** |
| NonambigQA | **64.2** | 42.2 | 54.0 | 65.1 | 69.0 | **69.8** | 69.4 | 74.0 | **77.9** |
| SciQ | **67.8** | 58.7 | 67.8 | 67.3 | 75.7 | **76.6** | 55.9 | **72.0** | 70.4 |
| TriviaQA | 69.5 | 51.6 | 66.3 | 70.2 | 72.5 | **74.6** | 64.2 | 71.4 | **75.3** |
| *Average* | **62.0** | 55.7 | 61.9 | 66.4 | 68.8 | **70.9** | 56.4 | **64.6** | 64.3 |

Table 4: AUROC comparison for VC, TL, VC+TL metrics for 7B models using the *linguistic confidence* prompt (i.e., asking the model to output the phrase which best expresses its confidence in the answer, see Prompt 1 in Appendix E). AUROC values are multiplied by 100 for readability. Best value for each model on each dataset is bolded.

| Dataset | Qwen2.5-7B-Instruct | | | Nemotron-7B | | | OpenThinker3-7B | | |
|---|---|---|---|---|---|---|---|---|---|
| | VC | TL | VC+TL | VC | TL | VC+TL | VC | TL | VC+TL |
| MedQA | 64.8 | 64.7 | **67.7** | 72.2 | **80.1** | 80.1 | 60.7 | 61.1 | **64.3** |
| SuperGPQA | **53.9** | 50.3 | 53.6 | **59.1** | 51.2 | 55.4 | **53.0** | 51.1 | 52.4 |
| Folktexts | 60.0 | 60.8 | **64.8** | 57.9 | **62.7** | 61.6 | 56.0 | 57.7 | **58.7** |
| MediQ | **63.1** | 56.9 | 62.6 | 67.6 | 67.5 | **69.3** | 60.0 | 56.5 | **60.7** |
| MedMCQA | 61.3 | 57.9 | **61.9** | 67.8 | 69.6 | **71.8** | 59.1 | 59.4 | **62.6** |
| MMLU | **63.2** | 56.4 | 62.8 | 70.9 | 77.6 | **80.0** | 67.1 | 68.7 | **73.8** |
| MMLU-Pro | | | | | | | | | |
| NoMath | 61.9 | 57.8 | **62.8** | 66.2 | 73.9 | **75.3** | 64.6 | 66.3 | **70.9** |
| NonambigQA | **58.4** | 43.9 | 48.8 | 66.0 | 68.0 | **70.0** | 70.9 | 65.0 | **71.4** |
| SciQ | 60.5 | 64.1 | **66.3** | 63.8 | 71.7 | **73.0** | 61.0 | 70.6 | **72.1** |
| TriviaQA | 58.2 | 57.0 | **60.5** | 72.3 | 72.6 | **75.2** | 67.8 | 68.8 | **73.3** |
| *Average* | 60.5 | 57.0 | **61.2** | 66.4 | 69.5 | **71.2** | 62.0 | 62.5 | **66.0** |

Table 5: AUROC comparison for VC, TL, VC+TL metrics for 7B models using the *numeric* confidence prompt (i.e., asking the model to output a confidence score in [0,100], see Prompt 2 in Appendix E). AUROC values are multiplied by 100 for readability. Best value for each model on each dataset is bolded.

| Dataset | Qwen2.5-7B-Instruct | | | Nemotron-7B | | | OpenThinker3-7B | | |
|---|---|---|---|---|---|---|---|---|---|
| | VC | TL | VC+TL | VC | TL | VC+TL | VC | TL | VC+TL |
| MedQA | 60.0 | 57.3 | **62.6** | 65.1 | 75.9 | **77.6** | 59.7 | 61.0 | **64.5** |
| SuperGPQA | **54.2** | 49.5 | 52.9 | **56.9** | 51.2 | 53.4 | **57.4** | 50.9 | 55.9 |
| Folktexts | **54.0** | 42.8 | 49.2 | **66.2** | 53.5 | 62.6 | 51.3 | **54.5** | 54.1 |
| MediQ | 53.8 | 51.0 | **54.2** | 67.9 | 68.0 | **71.6** | 58.4 | 55.2 | **59.0** |
| MedMCQA | 60.3 | 57.4 | **62.9** | 64.7 | 64.6 | **68.2** | **65.9** | 57.1 | 64.4 |
| MMLU | 60.1 | 59.1 | **67.0** | 66.9 | 68.4 | **73.8** | **72.8** | 63.3 | 72.6 |
| MMLU-Pro | | | | | | | | | |
| NoMath | **64.2** | 53.0 | 64.0 | 62.3 | 59.8 | **63.0** | 64.9 | 61.2 | **65.8** |
| NonambigQA | 67.8 | 64.7 | **70.3** | 59.0 | 61.3 | **62.7** | **75.7** | 65.2 | 72.4 |
| SciQ | 69.4 | 63.1 | **71.5** | 61.9 | 59.5 | **64.0** | 61.3 | 61.2 | **63.1** |
| TriviaQA | 73.0 | 72.3 | **79.8** | 64.1 | 62.9 | **66.5** | 70.4 | 66.6 | **72.2** |
| *Average* | 61.7 | 57.0 | **63.4** | 63.5 | 62.5 | **66.3** | 63.8 | 59.6 | **64.4** |

Table 6: AUROC comparison for VC, TL, VC+TL metrics for 7B models using the *top-k* prompt (i.e., asking the model to output the top-k most likely answers then choosing the one with the highest confidence score; see Prompt 3 in Appendix E). AUROC values are multiplied by 100 for readability. Best value for each model on each dataset is bolded.

## F.3   USEFULNESS OF TRACE LENGTH WITHOUT VERBAL CONFIDENCE PROMPT

| Dataset | OpenThinker2-32B | | Skywork-OR1-32B | | R1-Distill-32B | |
|---|---|---|---|---|---|---|
| | Numeric TL | Answer-only TL | Numeric TL | Answer-only TL | Numeric TL | Answer-only TL |
| MedQA | 78.4 | **79.4** | 80.9 | **81.6** | **75.9** | 75.9 |
| SuperGPQA | 58.4 | **61.0** | 57.1 | **58.7** | **60.1** | 56.5 |
| Folktexts | **64.1** | 61.2 | 56.4 | **59.8** | **62.5** | 60.3 |
| MediQ | 70.2 | **71.6** | 70.5 | **71.3** | **67.0** | 66.8 |
| MedMCQA | 71.5 | **71.8** | **71.5** | 68.9 | 64.5 | **69.6** |
| MMLU | 81.6 | **82.5** | 78.8 | **80.3** | **72.2** | 72.1 |
| MMLU-Pro NoMath | **80.2** | 78.2 | **77.6** | 75.8 | 76.4 | **78.5** |
| NonambigQA | 73.2 | **73.7** | **72.9** | 68.1 | **66.8** | 64.1 |
| SciQ | **75.8** | 75.1 | 76.0 | **77.0** | **66.7** | 62.9 |
| TriviaQA | **84.1** | 80.6 | **79.1** | 78.9 | 72.6 | **73.5** |
| *Average* | **73.8** | 73.5 | **72.1** | 72.0 | **68.5** | 68.0 |

Table 7: Comparison between the AUROC of trace length for the numeric confidence Prompt 2 and answer-only Prompt 4 (32B models). AUROC values are multiplied by 100 for readability. Even when asking for only the answer, and not the verbal confidence, trace length still predicts correctness.

| Dataset | Nemotron-7B | | OpenThinker3-7B | |
|---|---|---|---|---|
| | Numeric TL | Answer-only TL | Numeric TL | Answer-only TL |
| MedQA | **80.1** | 76.5 | 61.1 | **62.3** |
| SuperGPQA | **51.2** | 50.1 | 51.1 | **54.7** |
| Folktexts | 62.7 | **65.7** | 57.7 | **62.5** |
| MediQ | 67.5 | **68.2** | 56.5 | **58.9** |
| MedMCQA | **69.6** | 67.7 | 59.4 | **63.3** |
| MMLU | **77.6** | 70.1 | **68.7** | 66.8 |
| MMLU-Pro NoMath | **73.9** | 72.0 | 66.3 | **67.2** |
| NonambigQA | **68.0** | 62.3 | **65.0** | 63.6 |
| SciQ | 71.7 | **75.1** | **70.6** | 65.5 |
| TriviaQA | **72.6** | 70.5 | **68.8** | 68.5 |
| *Average* | **69.5** | 67.8 | 62.5 | **63.3** |

Table 8: Comparison between the AUROC of trace length for the numeric confidence Prompt 2 and answer-only prompts Prompt 4 (7B models). AUROC values are multiplied by 100 for readability. Even when asking for only the answer, and not the verbal confidence, trace length still predicts correctness.

## F.4   ACCURACY AND BRIER SCORE OF 32B MODELS

| Dataset | Qwen2.5-Instruct | | iw-SFT | | OpenThinker2 | | Skywork-OR1 | | R1-Distill | |
|---|---|---|---|---|---|---|---|---|---|---|
| | Acc | Brier | Acc | Brier | Acc | Brier | Acc | Brier | Acc | Brier |
| MedQA | 75.9 | 19.1 | 84.2 | 11.2 | 84.2 | 12.5 | 84.6 | 11.1 | 83.0 | 13.3 |
| SuperGPQA | 37.5 | 49.1 | 45.7 | 36.0 | 45.6 | 37.9 | 44.3 | 37.9 | 39.1 | 44.8 |
| Folktexts | 73.1 | 18.6 | 75.3 | 16.7 | 71.9 | 19.6 | 71.0 | 19.0 | 70.8 | 19.2 |
| MediQ | 51.3 | 30.7 | 60.6 | 24.9 | 62.1 | 26.7 | 61.2 | 25.6 | 59.6 | 28.0 |
| MedMCQA | 62.8 | 29.0 | 67.1 | 22.3 | 66.8 | 23.7 | 68.5 | 22.3 | 65.7 | 25.5 |
| MMLU | 87.1 | 10.8 | 89.9 | 8.0 | 90.4 | 7.6 | 89.5 | 7.9 | 88.1 | 9.7 |
| MMLU-Pro NoMath | 67.4 | 26.6 | 71.8 | 20.0 | 73.4 | 20.5 | 73.7 | 19.5 | 71.3 | 22.9 |
| NonambigQA | 54.6 | 37.6 | 54.1 | 27.9 | 53.8 | 32.0 | 52.9 | 31.3 | 51.4 | 36.3 |
| SciQ | 81.1 | 17.2 | 79.5 | 15.5 | 83.9 | 13.1 | 86.8 | 10.3 | 84.3 | 13.3 |
| TriviaQA | 74.5 | 21.0 | 79.5 | 13.6 | 78.1 | 15.1 | 77.2 | 15.3 | 76.2 | 18.0 |
| *Average* | 66.5 | 26.0 | 70.8 | 19.6 | 71.0 | 20.9 | 71.0 | 20.0 | 69.0 | 23.1 |

Table 9: Accuracy and Brier Score (both values $\times 100$) comparison for 32B models using Prompt 2 (numeric confidence in $[0, 100]$).

## G  UTILITY OF EPISTEMIC MARKERS AS UNCERTAINTY SCORE

In this section, we demonstrate that the number of *epistemic markers* provide comparable performance to trace length for uncertainty quantification. The generation details are identical to Appendix F.1: namely, we evaluate with temperature 0 across four reasoning models and three prompts. Based on observing the most common "BestToken" from the tables in Figure 4 and Appendix I, we choose the following "representative set" of epistemic markers. We use the following regex list for epistemic markers, and use the total count of all markers as our uncertainty score "EM" (we ignore case when counting the number of markers).

```
# List of markers using regex parsing
epistemic_markers = [
    r'(maybe)\b',
    r'(perhaps)\b',
    r'(possibly)\b',
    r'(considering)\b',
    r'(however)\b',
    r'(or)\b',
]
```

Tables 10 and 11 demonstrate that across datasets and models, the AUROC of trace length (TL) and the number of epistemic markers (EM) are comparable for 32B models. We also include the AUROC of verbalized confidence (VC) for reference.

| Dataset | Qwen2.5-Instruct | | | iw-SFT | | | OpenThinker2 | | | Skywork-OR1 | | | R1-Distill | | |
|---|---|---|---|---|---|---|---|---|---|---|---|---|---|---|---|
| | TL | EM | VC | TL | EM | VC | TL | EM | VC | TL | EM | VC | TL | EM | VC |
| MedQA | 58.9 | 51.7 | **67.6** | **81.0** | 77.9 | 71.7 | **80.0** | 78.0 | 69.6 | **81.4** | 79.3 | 71.7 | **73.8** | 69.5 | 63.8 |
| SuperGPQA | 51.6 | 51.3 | **61.1** | 58.4 | 58.2 | **61.0** | 59.6 | **61.5** | 57.3 | 57.2 | **59.4** | 56.5 | 57.3 | **59.7** | 55.9 |
| Folktexts | **60.1** | 59.5 | 51.9 | **72.2** | 67.1 | 64.4 | **63.9** | 63.9 | 63.1 | **58.0** | 56.8 | 53.5 | 59.0 | 56.7 | **62.0** |
| MediQ | 56.9 | 56.2 | **66.8** | **71.4** | 70.2 | 63.0 | 69.7 | **69.9** | 65.1 | 66.7 | **67.9** | 62.7 | 65.7 | **65.8** | 60.1 |
| MedMCQA | 57.6 | 59.0 | **70.7** | **76.5** | 73.1 | 69.3 | **71.3** | 70.2 | 66.5 | 67.3 | **69.1** | 65.6 | **67.0** | 65.7 | 66.1 |
| MMLU | 59.4 | 56.7 | **73.0** | **82.9** | 82.2 | 78.9 | **82.5** | 81.4 | 76.4 | 76.9 | **78.4** | 71.8 | 70.0 | **72.3** | 67.5 |
| MMLU-Pro | | | | | | | | | | | | | | | |
| NoMath | 57.3 | 59.7 | **69.8** | **79.0** | 74.5 | 71.2 | **79.1** | 75.8 | 70.6 | **76.8** | 74.3 | 67.9 | **76.2** | 73.0 | 62.9 |
| NonambigQA | 51.3 | 60.6 | **72.1** | 74.5 | **76.3** | 73.3 | 73.9 | **75.1** | 73.1 | 72.6 | **76.5** | 72.6 | 67.8 | 71.8 | **74.2** |
| SciQ | 58.7 | 61.1 | **68.9** | **76.0** | 74.3 | 75.2 | **77.9** | 75.7 | 71.2 | 72.3 | 71.9 | **72.6** | 66.7 | 66.1 | **69.7** |
| TriviaQA | 60.8 | 64.7 | **77.4** | **84.5** | 83.9 | 82.2 | **85.1** | 82.9 | 78.1 | 80.3 | 80.0 | **81.0** | 75.1 | 76.3 | **78.5** |
| *Average* | 57.3 | 58.1 | **67.9** | **75.6** | 73.8 | 71.0 | **74.3** | 73.4 | 69.1 | 70.9 | **71.3** | 67.6 | **67.9** | 67.7 | 66.1 |

Table 10: AUROC comparison for TL, EM, VC metrics for 32B models using the *linguistic confidence* prompt (i.e., asking the model to output the phrase which best expresses its confidence in the answer, see Prompt 1 in Appendix E). AUROC values are multiplied by 100 for readability. Best value for each model on each dataset is bolded.

| Dataset | Qwen2.5-Instruct | | | iw-SFT | | | OpenThinker2 | | | Skywork-OR1 | | | R1-Distill | | |
|---|---|---|---|---|---|---|---|---|---|---|---|---|---|---|---|
| | TL | EM | VC | TL | EM | VC | TL | EM | VC | TL | EM | VC | TL | EM | VC |
| MedQA | 60.7 | 54.1 | **71.4** | **80.1** | 75.4 | 78.5 | **78.4** | 75.2 | 73.9 | 80.9 | 79.4 | **82.4** | 75.9 | 70.1 | **77.2** |
| SuperGPQA | 51.7 | 53.0 | **57.3** | 57.8 | 57.6 | **59.8** | 58.4 | 60.1 | **62.2** | 57.1 | 59.0 | **60.6** | 60.1 | 61.2 | **62.7** |
| Folktexts | 62.6 | 61.1 | **68.1** | 69.3 | 65.8 | **72.4** | 64.1 | 63.3 | **68.4** | 56.4 | 56.9 | **72.8** | 62.5 | 60.8 | **72.8** |
| MediQ | 58.7 | 58.2 | **66.9** | 70.1 | 68.6 | **70.6** | 70.2 | **70.7** | 65.8 | 70.5 | **71.0** | 70.9 | 67.0 | **67.8** | 67.7 |
| MedMCQA | 62.4 | 58.3 | **69.9** | **72.0** | 70.3 | 71.1 | 71.5 | 69.9 | **71.7** | 71.5 | 71.0 | **72.0** | 64.5 | 63.9 | **71.1** |
| MMLU | 65.3 | 61.1 | **72.0** | **82.2** | 80.8 | 78.4 | 81.6 | **83.3** | 79.2 | 78.8 | 79.9 | **83.2** | 72.2 | 72.2 | **76.1** |
| MMLU-Pro | | | | | | | | | | | | | | | |
| NoMath | 60.1 | 59.6 | **71.1** | **77.4** | 72.2 | 75.9 | **80.2** | 78.0 | 73.2 | **77.6** | 74.8 | 73.9 | **76.4** | 72.1 | 69.3 |
| NonambigQA | 53.1 | 60.7 | **66.7** | 73.7 | 74.8 | **79.0** | 73.2 | 74.4 | **77.9** | 72.9 | 76.5 | **80.0** | 66.8 | 71.0 | **75.7** |
| SciQ | 62.8 | 62.7 | **67.3** | 74.5 | 73.5 | **77.7** | **75.8** | 74.8 | 73.3 | 76.0 | 75.8 | **79.8** | 66.7 | 65.0 | **71.8** |
| TriviaQA | 60.8 | 64.9 | **72.8** | 81.3 | 80.3 | **81.4** | 84.1 | 83.6 | **86.2** | 79.1 | 80.8 | **84.1** | 72.6 | 75.2 | **79.3** |
| *Average* | 59.8 | 59.4 | **68.4** | 73.8 | 71.9 | **74.5** | **73.8** | 73.3 | 73.2 | 72.1 | 72.5 | **76.0** | 68.5 | 67.9 | **72.4** |

Table 11: AUROC comparison for TL, EM, VC metrics for 32B models using the *numeric* confidence prompt (i.e., asking the model to output a confidence score in [0,100], see Prompt 2 in Appendix E). AUROC values are multiplied by 100 for readability. Best value for each model on each dataset is bolded.

# H  LIST OF HIGHEST ENTROPY TOKENS

In this section, we list the highest entropy tokens for Qwen and Skywork when averaged over all ten datasets. Prompt used elicits only the answer and not verbal confidence (Prompt 4). Common epistemic markers often used to examine uncertainty are highlighted. Only includes tokens which appear in at least three datasets, and twenty responses per dataset.

| Skywork-OR1-32B | | | Qwen2.5-32B-Instruct | | |
|---|---|---|---|---|---|
| **Rank** | **Token** | **Avg Entropy** | **Rank** | **Token** | **Avg Entropy** |
| 1 | —if | 1.449 | 1 | Among | 2.321 |
| 2 | Are | 1.430 | 2 | Considering | 2.268 |
| 3 | **Could** | 1.417 | 3 | Based | 2.266 |
| 4 | **Sometimes** | 1.408 | 4 | possibly | 2.193 |
| 5 | Usually | 1.401 | 5 | focuses | 2.146 |
| 6 | outline | 1.383 | 6 | lack | 2.146 |
| 7 | Yeah | 1.362 | 7 | typical | 2.136 |
| 8 | yeah | 1.354 | 8 | Given | 2.130 |
| 9 | Whereas | 1.329 | 9 | Therefore | 2.118 |
| 10 | Well | 1.314 | 10 | thus | 2.107 |
| 11 | **Probably** | 1.301 | 11 | **sometimes** | 2.101 |
| 12 | Another | 1.296 | 12 | unless | 2.095 |
| 13 | Plus | 1.293 | 13 | implies | 2.094 |
| 14 | Also | 1.287 | 14 | **potentially** | 2.091 |
| 15 | What | 1.283 | 15 | Thus | 2.070 |
| 16 | clarify | 1.275 | 16 | broader | 2.067 |
| 17 | Oh | 1.272 | 17 | analyzing | 2.064 |
| 18 | putting | 1.249 | 18 | suggesting | 2.062 |
| 19 | considering | 1.239 | 19 | mentions | 2.052 |
| 20 | **Perhaps** | 1.232 | 20 | conclude | 2.051 |
| 21 | let | 1.230 | 21 | provides | 2.048 |
| 22 | referring | 1.225 | 22 | considering | 2.048 |
| 23 | Typically | 1.223 | 23 | suggest | 2.046 |
| 24 | recap | 1.218 | 24 | correctly | 2.045 |
| 25 | tends | 1.217 | 25 | suggests | 2.041 |
| 26 | another | 1.210 | 26 | Since | 2.037 |
| 27 | mis | 1.209 | 27 | leads | 2.019 |
| 28 | More | 1.207 | 28 | Now | 2.014 |
| 29 | Those | 1.207 | 29 | implications | 2.012 |
| 30 | Some | 1.206 | 30 | indicates | 2.004 |
| 31 | ... | 1.201 | 31 | since | 1.990 |
| 32 | Does | 1.195 | 32 | make | 1.988 |
| 33 | **wait** | 1.190 | 33 | evaluate | 1.987 |
| 34 | sometimes | 1.190 | 34 | align | 1.986 |
| 35 | Alright | 1.187 | 35 | Let | 1.981 |
| 36 | simpler | 1.182 | 36 | clinical | 1.978 |
| 37 | How | 1.175 | 37 | contribute | 1.976 |
| 38 | Unless | 1.175 | 38 | useful | 1.971 |
| 39 | Not | 1.172 | 39 | among | 1.967 |
| 40 | Common | 1.170 | 40 | usually | 1.960 |
| 41 | Did | 1.170 | 41 | aligned | 1.957 |
| 42 | quite | 1.165 | 42 | relate | 1.953 |
| 43 | **maybe** | 1.159 | 43 | reasonable | 1.950 |
| 44 | going | 1.159 | 44 | indicating | 1.950 |
| 45 | Like | 1.155 | 45 | often | 1.947 |
| 46 | Without | 1.155 | 46 | imply | 1.946 |
| 47 | One | 1.152 | 47 | critical | 1.939 |
| 48 | People | 1.152 | 48 | typically | 1.937 |
| 49 | **Maybe** | 1.151 | 49 | makes | 1.937 |
| 50 | extremely | 1.149 | 50 | check | 1.934 |

Table 12: Top 50 highest entropy tokens for Skywork-OR1-32B and Qwen2.5-32B-Instruct averaged across all datasets with Prompt 4.

# I  ADDITIONAL DETAILS ON FORKING TOKEN PLOTS AND TABLES

In this section, we provide additional plots similar to Figure 5 and tables similar to the right of Figure 4. First, we describe how we generate high entropy forking tokens. We evaluate three models — OpenThinker2-32B, Skywork-OR1, and Qwen2.5-32B-Instruct — at temperature 1 with generation length 8192 and Prompt 4. To approximate the entropy of each token, at generation time we compute the entropy of the distribution over the top 30 highest probability tokens using the logprobs returned by vLLM. This is for efficiency reasons — computing the entropy over the entire vocabulary is too computationally expensive. However, we qualitatively observed that this captured most of the mass in the entire distribution.

We compute the 50 tokens with the highest *average* entropy for each model on each dataset. We only include tokens which appear in at least 20 separate responses in each dataset — this rules out tokens which are simply possible answers to particular questions. We call these 50 tokens the set of "forking tokens" (FT) for a particular model on a particular dataset. When we say that we use FT as an uncertainty score, we mean that we count the number of times that any of the tokens in FT appear in a reasoning trace. The higher this number, the less likely a generation is of producing a correct response.

In the right side of Figure 4, we demonstrated that for OpenThinker2-32B, forking tokens (FT) perform similarly to trace length in terms of AUROC. We also demonstrated a "best forking token" (BFT), which is the single token among the 50 which provides best AUROC, and provided the value of that token in "BestToken". The following two tables are analogous to Figure 4, but for the base Qwen2.5 model and the Skywork reasoning model. These tables demonstrate that, like length, forking tokens become much more useful after reasoning post-training.

| Dataset | TL | FT | TL+FT | SP | BFT | BestToken |
|---|---|---|---|---|---|---|
| Folktexts | 55.4 | 55.8 | **56.5** | 55.0 | 54.2 | Considering |
| MediQ | 57.0 | 57.7 | 58.6 | **60.7** | 57.3 | could |
| MMLU-Pro NoMath | 60.1 | 57.5 | 60.3 | **63.2** | 57.8 | Given |
| MedMCQA | 57.4 | 59.7 | 60.2 | **61.9** | 57.6 | would |
| MedQA | 60.0 | 58.6 | 61.1 | **65.4** | 56.8 | could |
| MMLU | 60.6 | 61.1 | 64.7 | **67.7** | 59.5 | might |
| SciQ | 62.1 | 56.0 | 60.0 | **63.1** | 54.8 | often |
| SuperGPQA | 53.5 | 54.8 | 55.8 | **56.6** | 53.2 | seems |
| TriviaQA | 69.7 | 65.2 | 68.0 | **72.9** | 67.7 | However |
| NonambigQA | 58.9 | 60.0 | 60.7 | **63.9** | 60.2 | However |
| *Average* | *59.5* | *58.6* | *60.6* | ***63.0*** | *57.9* | *-* |

Figure 12: **Qwen2.5-32B-Instruct.** Performance of trace length (TL), forking tokens (FT), TL+FT, Sequence Probability (SP), and best forking token (BFT). The single best forking token for each dataset is also given in "BestToken"

| Dataset | TL | FT | TL+FT | SP | BFT | BestToken |
|---|---|---|---|---|---|---|
| Folktexts | 54.1 | 56.7 | 56.2 | 56.1 | **61.7** | Considering |
| MediQ | 70.5 | 70.7 | **71.1** | 70.8 | 69.7 | perhaps |
| MMLU-Pro NoMath | 76.8 | 75.5 | 77.1 | **77.8** | 76.7 | perhaps |
| MedMCQA | 66.8 | 67.0 | 67.2 | **68.6** | 67.8 | maybe |
| MedQA | 78.1 | 78.1 | 78.5 | **79.1** | 79.1 | perhaps |
| MMLU | 72.7 | 72.5 | 72.8 | 74.2 | **76.4** | perhaps |
| SciQ | 69.6 | 68.4 | 69.4 | **70.9** | 67.9 | perhaps |
| SuperGPQA | 55.8 | 58.5 | 57.3 | 55.9 | **60.0** | perhaps |
| TriviaQA | 78.5 | 80.1 | 80.0 | **82.9** | 78.7 | maybe |
| NonambigQA | 70.4 | 71.3 | 72.1 | **74.3** | 64.1 | Or |
| *Average* | *69.3* | *69.9* | *70.2* | ***71.1*** | *70.2* | *-* |

Figure 13: **Skywork-Or1-32B.** Performance of trace length (TL), forking tokens (FT), TL+FT, Sequence Probability (SP), and best forking token (BFT). The single best forking token for each dataset is also given in "BestToken".

In the main text, Figure 5 demonstrates the utility of adding each forking token one by one to a "working set" in terms of the AUROC. We include more detailed version of the plots here, for a variety of datasets and models. There are a few important quantities in each of the following plots:

- The blue line shows the cumulative AUROC of using the total count of the top $k$ highest tokens in the generated traces in order to predict correctness.

- The five tokens which lead to the largest "jump" in AUROC are highlighted and displayed on the plot.

- The purple line plots the performance of a greedy heuristic which maintains a working set of forking tokens, and greedily adds the forking token to the set which increases the AUROC by a maximum amount.

- The dashed green line represents the AUROC of using the sequence probability as an uncertainty score — this is simply the AUROC of using the sum of all top token logprobs as the uncertainty score.

- The yellow line demonstrates the AUROC of using the total summed entropy across the generation.

- Finally, the red dashed line is simply the AUROC of the reasoning trace length (in tokens).

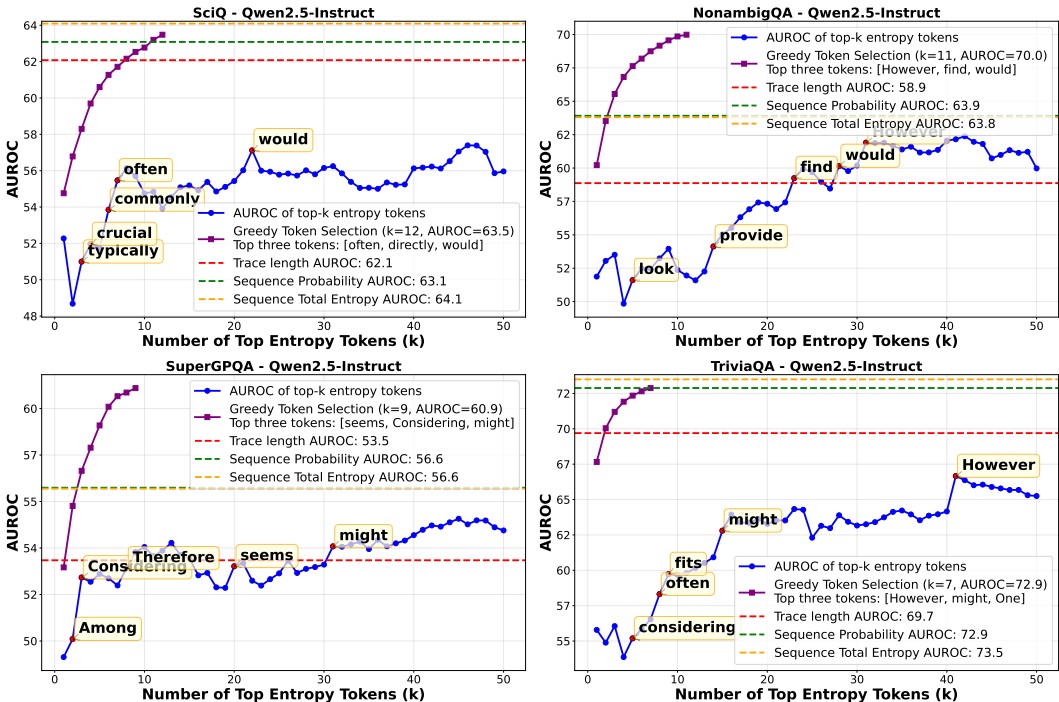

Figure 14: **Forking token plots for Qwen2.5-32B-Instruct.** See Appendix I for detailed description of quantities plotted. The utility of adding forking tokens to the "working set" of the top $k$ highest entropy tokens is roughly monotonic, but not as clear for the Qwen base model as in the reasoning models below.

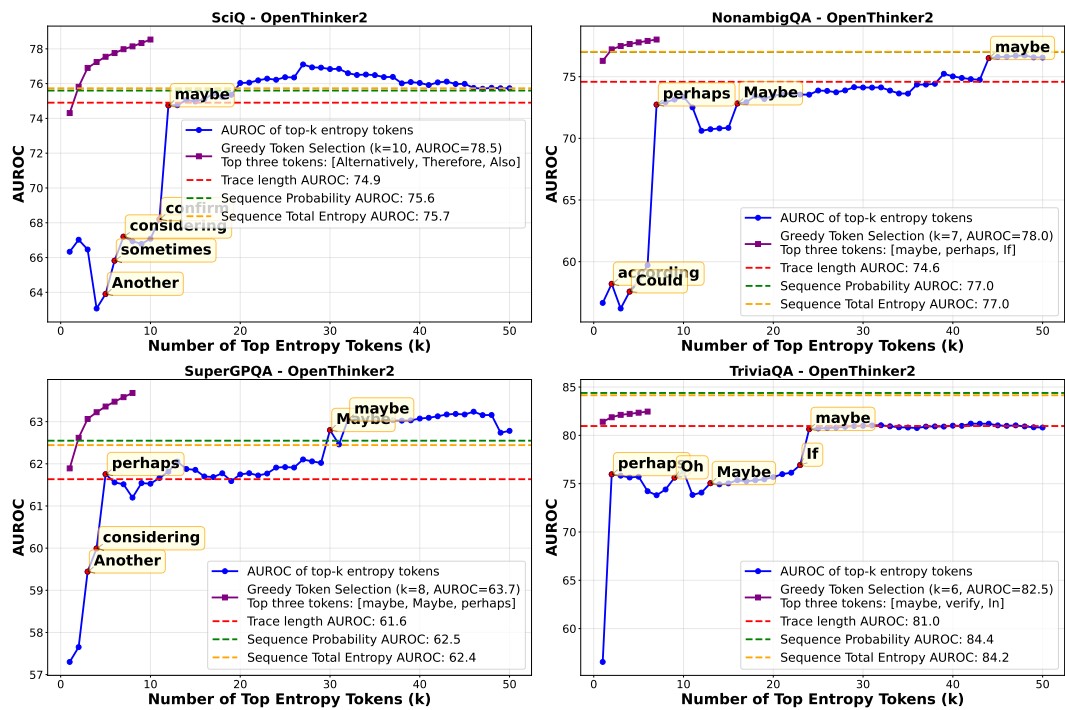

Figure 15: **Forking token plots for OpenThinker2-32B.** See Appendix I for detailed description of quantities plotted. The utility of adding forking tokens to the "working set" of the top $k$ highest entropy tokens is roughly monotonic. In addition, there often exists a *single* token (the first position of the purple greedy token selection line) which performs as well as trace length in terms of AUROC.

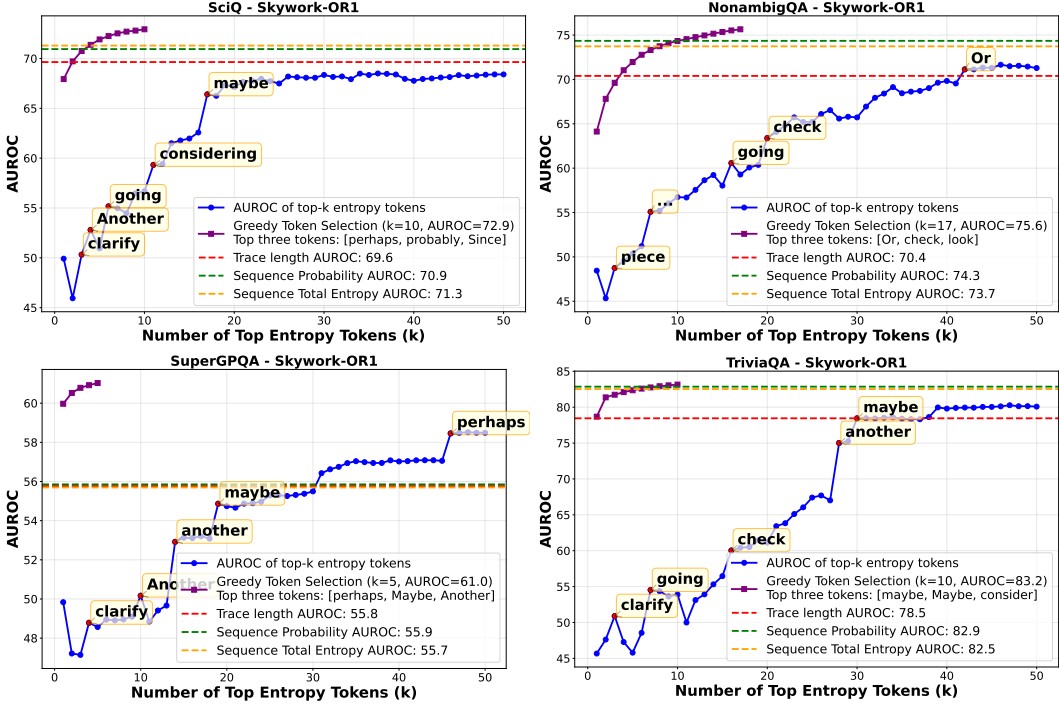

Figure 16: **Forking token plots for Skywork-OR1-32B.** See Appendix I for detailed description of quantities plotted. The utility of adding forking tokens to the "working set" of the top $k$ highest entropy tokens is roughly monotonic. In addition, there often exists a *single* token (the first position of the purple greedy token selection line) which performs as well as trace length in terms of AUROC.

## J  SKYWORK-OR1 LENGTH DISTRIBUTION

In this section, we present Figure 17, which demonstrates that Skywork-OR1 still has a length bias for correct and incorrect responses. This holds even though OR1 was post-trained via a variant of GRPO with the length bias term removed (He et al., 2025a, Section 3). We note that OR1 is trained on top of R1-Distill, which itself was trained on reasoning traces of R1. Therefore, the length bias of OR1 could have emerged from the prior steps used to obtain R1. Nonetheless, it is difficult to find open source 32B reasoning models which explicitly compare GRPO vs. Dr. GRPO (or other variants of GRPO which seek to address the length bias of incorrect responses).

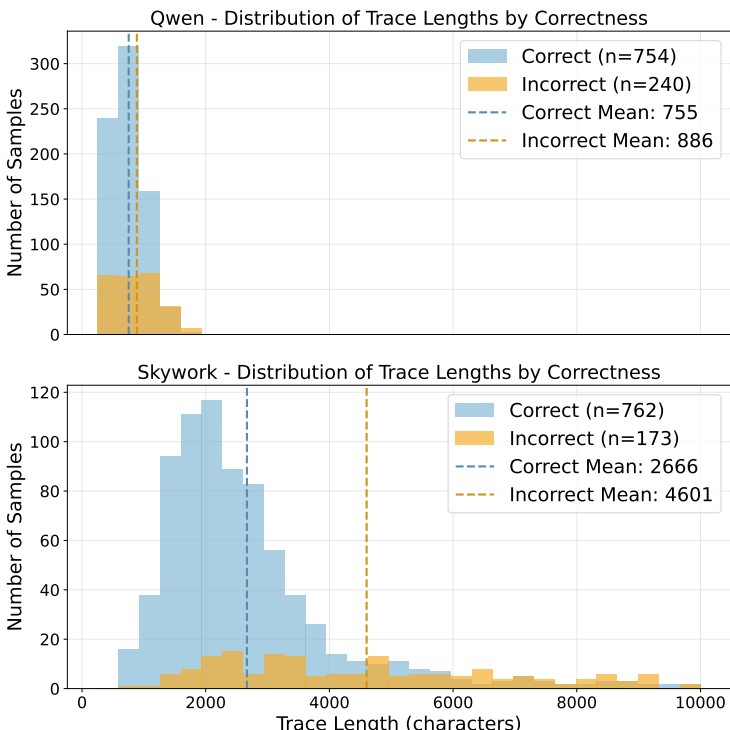

Figure 17: **Distribution of reasoning trace lengths in Qwen2.5-32B-Instruct and Skywork-OR1 on TriviaQA.** Mean length of correct and incorrect responses stretches far apart for Skywork-OR1, while the base Qwen model has them much closer together. Note that the accuracy of Qwen and OR1 are similar; the bottom histogram is missing incorrect samples because we cut the graph off at 10k characters for readability. Any additional incorrect samples can only spread the means further apart.

We note that Intellect-2 (Team et al., 2025b) and L1 (Aggarwal & Welleck, 2025) are both open source models which are post-trained to explicitly control reasoning trace length; we are instead interested in the emergence of length from standard reinforcement learning with verifiable 0-1 rewards, i.e., rewards or reinforcement learning algorithms which *do not* explicitly try to control or decrease the generated trace length. There are a variety of algorithms which do attempt to control trance length, such as GFPO (Shrivastava et al., 2025). Yet other approaches look to find a curriculum which slowly increases reasoning trace length through training (Luo et al., 2025).

## K    CORRELATION OF ACCURACY AND AUROC OF TL AND VC

In this section, we show that the AUROC of both trace length (TL) and verbal confidence (VC) are correlated with the underlying accuracy of the model. This echoes results from Mei et al. (2025) and Vanhoyweghen et al. (2025), who broadly show decreasing usefulness of UQ methods when accuracy is lowered.

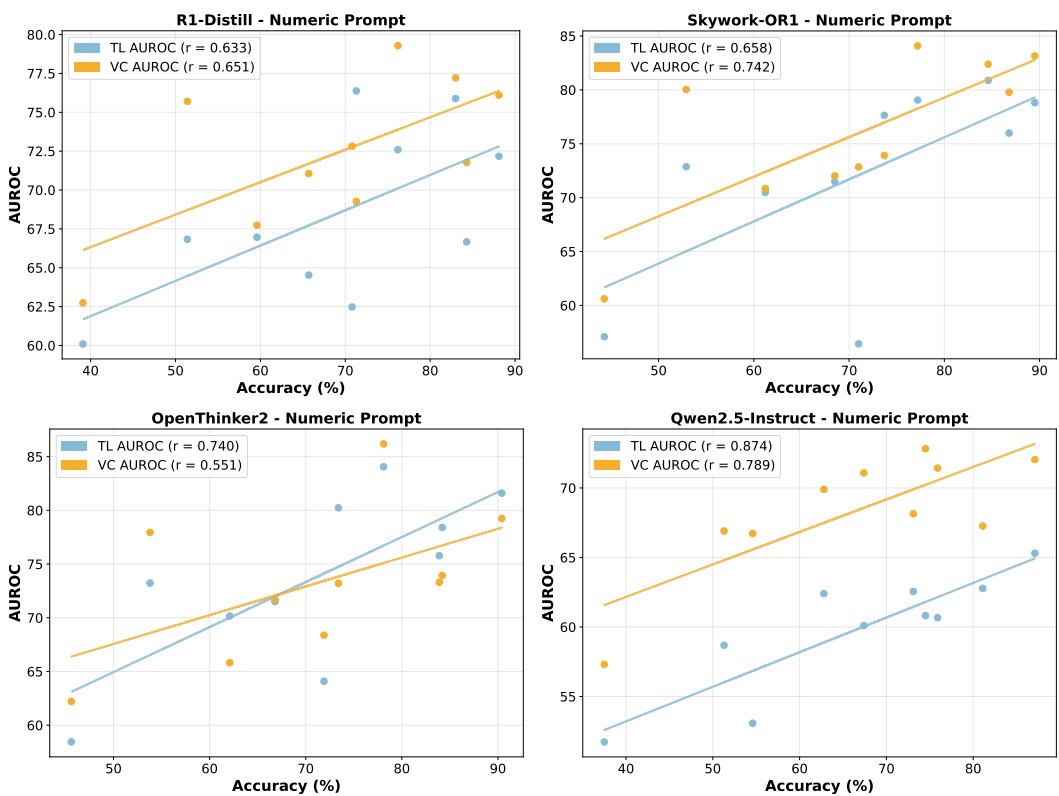

Figure 18: Correlations of trace length (TL) and verbal confidence (VC) AUROC with Accuracy across all ten datasets (using Prompt 2).

## L    GENERALIZATION TO NON-QWEN MODELS

Our main experiments focus exclusively on reasoning models that use Qwen2.5 as a base model. This restriction was necessary to allow us to directly compare uncertainty metrics both before and after reasoning training by accessing both the base model and its post-training counterpart. However, this narrow scope raises the question of whether the observed increase in trace length performance generalizes across different base models, or is unique to Qwen 2.5.

In this section, we give evidence that trace length is still a useful uncertainty estimate in three other reasoning models built on base models that are *not* Qwen2.5: GPT-OSS-20B, Phi-4-Reasoning-Plus (14B), and GLM-4.5-Air (106B). This supports our hypothesis that the utility of trace length as an uncertainty signal is a consequence of the RL post-training process rather than pathologies specific to any particular base model. We note that even though these are open-weight models, we have no access to the code, data, or algorithms used to generate them.

We run all experiments for this section with temperature 1.0 sampling via the OpenRouter API, with the reasoning budget set to 4096 tokens. For simplicity, we utilize Prompt 2 for all queries. The AUROC of verbal confidence (VC), trace length (TL), and their sum is presented in Table 13. The corresponding accuracies and Brier scores are in Table 15.

Together, these results demonstrate that trace length remains a useful uncertainty metric even for models built on non-Qwen base architectures, supporting the generality of our findings across different training procedures and model families.

| Dataset | Phi-4-Reasoning-Plus | | | Hermes-70B | | | GPT-OSS-20B | | | GLM-4.5-Air | | |
|---------|------|------|-------|------|------|-------|------|------|-------|------|------|-------|
| | VC | TL | VC+TL | VC | TL | VC+TL | VC | TL | VC+TL | VC | TL | VC+TL |
| NonambigQA | **58.9** | 51.9 | 55.3 | 69.0 | 72.9 | **75.2** | 84.9 | 84.4 | **86.3** | 72.0 | 65.6 | **73.0** |
| MedQA | 62.5 | 62.8 | **65.9** | 71.5 | 76.0 | **80.0** | 68.0 | 70.1 | **71.3** | 70.1 | 69.4 | **72.0** |
| SuperGPQA | **58.2** | 52.5 | 56.4 | 60.2 | 57.6 | **61.7** | **62.2** | 55.8 | 60.6 | 56.2 | **68.8** | 43.8 |
| TriviaQA | 63.3 | 62.8 | **66.6** | 72.7 | 80.7 | **81.9** | **93.8** | 91.6 | 93.7 | 74.6 | 76.6 | **79.3** |
| *Average* | 60.7 | 57.5 | **61.0** | 68.3 | 71.8 | **74.7** | 77.2 | 75.5 | **78.0** | 68.2 | **70.1** | 67.0 |

Table 13: AUROC comparison for VC, TL, VC+TL metrics for OpenRouter API models. VC: Verbalized Confidence. TL: Reasoning Trace Length. VC + TL: zero-shot combination. AUROC values are multiplied by 100 for readability. Best value for each model on each dataset is bolded.

| Dataset | Phi-4-Reasoning-Plus | | | Hermes-70B | | | GPT-OSS-20B | | | GLM-4.5-Air | | |
|---------|------|------|-------|------|------|-------|------|------|-------|------|------|-------|
| | VC | TL | VC+TL | VC | TL | VC+TL | VC | TL | VC+TL | VC | TL | VC+TL |
| NonambigQA | 1.34 | 1.86 | 1.85 | 1.90 | 1.87 | 1.83 | 1.19 | 1.23 | 1.14 | 1.72 | 1.89 | 1.76 |
| MedQA | 1.85 | 2.06 | 2.05 | 2.59 | 2.46 | 2.19 | 2.42 | 2.59 | 2.50 | 2.54 | 2.44 | 2.37 |
| SuperGPQA | 1.81 | 1.97 | 1.98 | 1.84 | 1.88 | 1.87 | 2.02 | 2.01 | 1.99 | 1.98 | 1.85 | 1.92 |
| TriviaQA | 1.54 | 2.12 | 2.09 | 2.72 | 2.65 | 2.57 | 1.60 | 1.89 | 1.67 | 2.48 | 2.52 | 2.20 |
| *Average* | 1.63 | 2.00 | 1.99 | 2.26 | 2.22 | 2.11 | 1.81 | 1.93 | 1.83 | 2.18 | 2.18 | 2.07 |

Table 14: Bootstrap standard deviation (1000 resamples) for AUROC values (VC, TL, VC+TL metrics) for OpenRouter API models. Values are multiplied by 100 for readability. Lower values indicate more stable estimates.

| Dataset | Phi-4-Reasoning-Plus | | Hermes-70B | | GPT-OSS-20B | | GLM-4.5-Air | |
|---------|------|-------|------|-------|------|-------|------|-------|
| | Acc | Brier | Acc | Brier | Acc | Brier | Acc | Brier |
| NonambigQA | 50.0 | 47.7 | 76.0 | 19.9 | 43.9 | 28.8 | 69.3 | 25.0 |
| MedQA | 80.4 | 21.7 | 89.5 | 8.9 | 80.6 | 11.5 | 86.1 | 13.6 |
| SuperGPQA | 33.2 | 60.5 | 43.3 | 42.7 | 33.4 | 43.7 | 40.0 | 49.9 |
| TriviaQA | 72.8 | 26.8 | 90.7 | 7.6 | 71.7 | 12.9 | 90.1 | 12.7 |

Table 15: Accuracy and Brier Score comparison for OpenRouter API models. Accuracy values are percentages (higher is better). Brier scores are multiplied by 100 (lower is better).

# M  RELIABILITY OF FORKING TOKENS RESULTS

In Section 5.1, Figure 5, and Appendix I, we computed the set of high entropy "forking tokens" on 1k examples from several datasets, and then evaluated the usefulness of forking token counts as a confidence signal on those same datasets. We provide two reliability checks for this approach to ensure that we are not overfitting on both the datasets and models we tested.

**Computing forking tokens on subsamples.**  To assess stability across different example sets, we re-computed forking tokens for OpenThinker2-32B using randomly drawn subsets of size 800 examples from the full 1k dataset. We repeated this proceedure ten times per dataset and measured the similarity between token rankings using Kendall's $\tau$. The results in Table 16 show average $\tau$ values around .8 ($\tau$ always lies in $[-1, 1]$), indicating that the ranking of top forking tokens is reasonably robust to perturbations in the example set. Furthermore, nearly 80% of the 50 forking tokens identified using the full 1K dataset of examples remain in the top 50 when using only random 800-example subsets.

These findings suggest that our results would remain largely stable under a validation/test split.

| Dataset | Mean $\tau$ | Std $\tau$ | Overlap | Overlap % | Std Overlap % |
|---|---|---|---|---|---|
| medmcqa | 0.826 | 0.045 | 40.9/50 | 81.8% | 3.6% |
| SuperGPQA | 0.821 | 0.028 | 42.7/50 | 85.4% | 3.1% |
| folktexts | 0.816 | 0.026 | 42.5/50 | 85.0% | 2.6% |
| mediQ | 0.813 | 0.027 | 44.0/50 | 88.0% | 2.2% |
| MedQA-USMLE-4-options | 0.794 | 0.058 | 42.0/50 | 84.0% | 2.0% |
| nonambigqa_val_1k | 0.785 | 0.041 | 38.2/50 | 76.4% | 1.5% |
| sciq | 0.782 | 0.044 | 43.2/50 | 86.4% | 2.0% |
| mmlu-pro-nomath | 0.775 | 0.043 | 40.1/50 | 80.2% | 3.4% |
| mmlu | 0.762 | 0.043 | 40.8/50 | 81.6% | 1.7% |
| trivia_qa | 0.697 | 0.064 | 39.4/50 | 78.8% | 3.2% |
| **Overall** | **0.787** | **0.057** | **41.4/50** | **82.8%** | **4.3%** |

Table 16: Forking token validation results for OpenThinker2-32B. Rankings computed from random subsets of 800 examples are compared to the baseline ranking from all 1000 examples using Kendall's $\tau$ correlation. Overlap shows the number of common tokens in the top-50. Results averaged over 10 random subsets per dataset.

**Test/Validation split for GPT-OSS-20B on MMLU-Pro-nomath** To further validate our findings, we also run an experiment with an explicit validation/test split for GPT-OSS-20B on MMLU-Pro-nomath. The results are shown in Table 17. Each split contains 1K examples. We use the validation split to greedily select the five best forking tokens (using the same greedy method described in Appendix I), and evaluate their performance as a confidence signal on the held-out test split. For comparison, we also include zero-shot baselines—verbal confidence, trace length, and sequence probability—which do not require tuning on validation data.

We also report the performance of a fixed set of epistemic markers: "maybe", "perhaps", "possibly", "considering", "however", and "or". We selected these by inspecting forking tokens across Qwen-based models Appendix G. Notably, despite GPT-OSS-20B having a different base model and post-training procedure, these epistemic markers achieve performance comparable to trace length, suggesting some degree of cross-model generalization.

Together, these results demonstrate that the forking tokens are relatively stable on splits of the same dataset, and may even transfer across different model families.

| Metric | AUROC |
|---|---|
| Verbal Confidence (VC) | 0.757 |
| Trace Length (TL) | 0.703 |
| Sequence Probability (SP) | 0.734 |
| Optimized Forking Tokens (5) | 0.729 |
| Epistemic Markers | 0.692 |

Table 17: Various metrics for GPT-OSS-20B on MMLU-Pro-nomath on a fresh 1K example test set. Optimized forking tokens selected using a hold-out validation set. All other metrics are zero-shot and do not rely on selection via a hold-out validation set.

