# OpenReview forum: "Trace Length is a Simple Uncertainty Signal in Reasoning Models"
_ICLR.cc/2026/Conference — Submitted to ICLR 2026_

### Official Review · Reviewer_twQ6 · 2025-11-01

**Soundness:** 2
**Presentation:** 3
**Contribution:** 2
**Rating:** 4
**Confidence:** 3

**Summary:**

In this work, the authors analyze how reasoning trace length can serve as a zero-shot uncertainty measure for large reasoning models with extensive experiments. Empirical results demonstrate that after reasoning post-training, trace length correlates strongly with correctness.

**Strengths:**

1. This work presents a novel insight that a simple signal like trace length can serve as confidence estimator.
2. The authors investigate extensive evaluations across multiple models, datasets, and prompts.
3. The paper is well written with clarity in writing.

**Weaknesses:**

1. The core idea of this work lacks novelty. The notion that longer reasoning shows uncertainty is intuitive and quite straightforward. Therefore, the contribution of this work may come from its empirical validation.
2. Lines 141–142 need to be revised. Tao et al. (2025) show that linguistic VC performs better than token-probability-based UQ and numeric VC, but not better than multi-sampling approaches. They only compare with consistency rate and majority vote in the Appendix, and the results cannot conclude that linguistic VC outperforms these methods. Moreover, some SOTA methods, such as Semantic Entropy and Kernel Language Entropy, are missing.
3. Regarding evaluation metrics, I agree that “AUROC and ECE (or Brier) are not necessarily aligned,” and this is a common phenomenon. However, they provide two distinct viewpoints: one measures discrimination, and the other measures how truthfully the confidence score reflects the empirical accuracy rate. Hence, from my understanding, ECE and its variants are still useful in many scenarios. In practice, a good confidence metric should yield both good AUROC and good ECE.
4. I noticed that the authors use OpenThinker2 and iw-SFT, which are models trained from SFT checkpoints. Can we claim that SFT helps improve the reliability of TL and VC? Also, what about the performance of the SFT-only model? I would like to see a comparison among SFT-only, SFT+RL, and RL-only settings.

**Questions:**

1. In lines 321–322, how do the authors choose the top k (50) highest-entropy tokens? Is the selection based on the whole test set or using a calibration/development set? If it is the former, is that a fair approach?
2. What is the author's conclusion regarding difficulty? Can the authors show the Spearman correlation between TL and difficulty?

---

> ### Author Response · Authors · 2025-11-22
>
> We thank the reviewer for careful consideration of our submission, and for useful clarifying comments and questions.
>
> **W1**: “The core idea of this work lacks novelty. The notion that longer reasoning shows uncertainty is intuitive and quite straightforward. Therefore, the contribution of this work may come from its empirical validation.”
>
> We agree that the underlying idea is intuitive, and we do not aim to focus on its novelty. Rather, our focus is indeed on a robust empirical validation (as intuitive ideas still require validation) as well as on identifying potential underlying mechanisms.
>
> **W2**: “Lines 141–142 need to be revised.”
>
> We thank the reviewer for pointing out our misinterpretation of the results of Tao et al. (2025). We have revised lines 141-142 in our new draft, pointing out that VC only generally outperforms single-pass black-box UQ approaches, not multi-sample approaches (such as semantic entropy). We stress that our focus in this work is on zero-shot single-pass approaches as these are especially useful in practice.
>
> **W3**: Evaluation metrics
>
> The reason that we did not report ECE throughout the paper is based on observations we made when performing research related to verbal confidence estimation. In particular, we found datasets where the model would obtain low ECE (and hence, be supposedly well calibrated) only because the model had a strong bias to output one or two specific verbal confidence values.
>
> For example, in the left side of Figure 9 in appendix B.2, we found a reasoning model which achieves excellent ECE of 0.03 on the MediQ dataset. However, this is almost solely because it nearly always outputs “60” as its verbal confidence on every question, regardless of whether it answered that question correctly. For this specific dataset, it happens to achieve around 60% accuracy. However, on other datasets where it has similar behavior, the ECE is much worse than 0.03, since the accuracy does not happen to be near 60%.
>
> In Appendix B.2, we provide some additional evidence for why we do not believe that ECE is as meaningful a calibration measure as AUROC for verbal confidence specifically.
>
> **W4**: SFT vs. RL vs. SFT+RL
>
> We recognize that the models we test are a combination of SFT, RL, and SFT+RL. We discuss this in Appendix C. As the reviewer points out, OpenThinker and iw-SFT are both SFT-only models. All other models, including the new models we added in Appendix L (please refer to part 1 of our overall response), are likely combinations of SFT and RL. As the recent SoTA open-source (not open-weight) model Olmo 3 reasoning demonstrates (released Nov. 20), a good post-training pipeline includes a variety of stages, with both SFT and RL.
>
> Given the broad range of models we test, and the fact that trace length is comparable to verbal confidence for each, we believe that our results hold in some generality with respect to post-training pipelines.
>
> **Q1**: In lines 321–322, how do the authors choose the top k (50) highest-entropy tokens? Is the selection based on the whole test set or using a calibration/development set? If it is the former, is that a fair approach?
>
> We choose the top 50 highest entropy tokens based on the entire 1K example test set. It is true that selection based on a holdout set is a more statistically sound approach. However, in this case we do not expect the top 50 set to have been overfit to any substantial extent. Indeed, in the newly added Table 16 in the new appendix M, we demonstrate that randomly subsampling the data results in forking token sets with large overlap (~80%) and high Kendall Tau ranking distance to the forking token set over the entire dataset.
>
> We follow this up with an additional experiment on GPT-OSS-20B (Table 17), where we show that using a validation set to select forking tokens results in performance comparable to verbal confidence on a hold-out test set. In summary, we do not expect forking tokens to stay relatively stable given a large enough sample size (e.g., 1K examples that we use in the main paper).
>
> **Q2**: What is the author's conclusion regarding difficulty? Can the authors show the Spearman correlation between TL and difficulty?
>
> Our conclusion is that both TL and VC capture at least some signal about uncertainty beyond question difficulty as defined by DeepMath (lines 417-420). This is captured in the LHS of Figure 6, which demonstrates that TL and VC continue to have good AUROC even after controlling for difficulty. As for Spearman correlation, we have run an additional experiment and found that TL and difficulty have a Spearman correlation of around 0.3 (see line 417). We thank the reviewer for pointing this out.

---

> > ### Comment · Reviewer_twQ6 · 2025-11-26
> >
> > Thank you for your detailed responses! While I still have some doubts about ECE, most of my concerns have been addressed satisfactorily, and hence I would increase my score.

---

### Official Review · Reviewer_Qs7z · 2025-11-01

**Soundness:** 4
**Presentation:** 3
**Contribution:** 2
**Rating:** 4
**Confidence:** 4

**Summary:**

The paper investigates how the length of a model’s reasoning trace can serve as a proxy for uncertainty in reasoning-trained LLMs. By analyzing multiple reasoning models (mainly Qwen2.5-based) across ten benchmarks, the authors find that longer reasoning traces are generally associated with lower answer accuracy, indicating higher uncertainty. They compare trace length (TL) against verbalized confidence (VC) and show that TL alone, or combined with VC, can effectively predict model correctness using AUROC as the primary metric.

**Strengths:**

**Simplicity and Practicality:** The proposed uncertainty signal (trace length) is extremely easy to compute and requires no model modification or additional prompts.

**Systematic Evaluation:** The authors evaluate across multiple datasets and models, using both verbal and numerical confidence prompts.

**Interpretability:** The study connects trace length with token-level entropy (“forking tokens”), offering some insight into why longer reasoning may correspond to uncertainty.

**Complementarity:** TL complements verbalized confidence, suggesting possible hybrid approaches for model reliability estimation.

**Weaknesses:**

**Lack of Theoretical Depth:** The paper mainly presents empirical correlations without a solid theoretical explanation of why trace length reflects uncertainty.

**Limited Generalization:** All models are variants of Qwen2.5; the findings may not generalize to other architectures or non-reasoning tasks.

**Shallow Analysis:** Statistical validation and significance testing are minimal; differences between TL and VC are often small.

**No Downstream Demonstration:** The work stops at correlation analysis and does not explore how TL-based uncertainty estimation can improve real-world applications like answer filtering or self-consistency.

**Questions:**

How stable is the correlation between trace length and uncertainty across different reasoning styles or prompt formats (e.g., Tree-of-Thought vs. Chain-of-Thought)?

Could trace length be confounded by model verbosity rather than genuine uncertainty?

How would TL perform in creative or open-ended tasks where long outputs are not necessarily uncertain?

Can TL be combined with other uncertainty signals (e.g., entropy, logit variance) to build a more principled UQ framework?

---

> ### Author Response · Authors · 2025-11-22
> **Response to weaknesses**
>
> We thank the reviewer for the careful review and relevant questions.
>
> **W1**: “Lack of Theoretical Depth”.
> We would like to clarify that our main contribution is to empirically identify the usefulness of trace length as an uncertainty metric, as well as present its complementarity with other zero-shot measures such as verbal confidence. Although a precise theoretical characterization for why length emerges as useful is outside the scope of our work, we do provide a hypothesis based on the existence of high-entropy forking tokens (Lines 319-323). We demonstrate that the number of forking tokens is highly correlated with trace length (Figure 4). The prior work of Wang et al. (2025) demonstrates that the entropy of forking tokens is amplified during RL post-training with GRPO (or its variants). Therefore, we hypothesize that forking tokens can be understood as “exploration” steps within the models’ generations, and as such, if more exploration is required, the model is less likely to find the correct solution to the problem (Lines 314-318).
>
> More broadly, we believe that a formal mathematical characterization would be premature as we still lack a good model of the various factors influencing a model’s correctness. The goal of our work is to identify a robust empirical phenomenon and point to areas that merit further investigation (specifically, forking tokens).
>
> **W2**: “Limited Generalization: All models are variants of Qwen2.5; the findings may not generalize to other architectures or non-reasoning tasks.”
>
> We thank you and other reviewers for pointing this out --- please refer to part 1 of our overall response.
>
> With respect to non-reasoning tasks: we emphasize that we tested this finding on 10 datasets, some of which are not typically considered to be reasoning tasks. This includes, for example, the factual recall task TriviaQA, and a dataset / task with high aleatoric uncertainty: FolkTexts. For these two tasks, as well as with the NonAmbigQA task, the base Qwen2.5 model and the four derived reasoning models in the main paper all have similar accuracy (see Table 9 in Appendix F.4) and therefore, the “reasoning” of post-trained models does not help improve its performance in terms of correctness. However, the reasoning models have better UQ via verbal confidence or trace length.
>
>
> **W3**: “Shallow Analysis: Statistical validation and significance testing are minimal; differences between TL and VC are often small.”
>
> We do not claim that TL unilaterally outperforms VC, and we agree that trace length (TL) and verbal confidence (VC) often have similar performance. This is why we argue throughout our paper that TL emerges as a confidence signal with similar / comparable usefulness as VC. We mention this in Lines 15-17 in the abstract, Line 72, Lines 211-212, and more places throughout.
>
> We do, however, argue that VC and TL can often be combined to obtain slightly better results (this is denoted as VC+TL throughout our paper). We report these findings in the main paper in Figure 2, where we show the averaged performance of each method for each model. In Figure 2, we average over all 10 datasets and 3 prompts we tested per model. Since the scale of the AUROC is incomparable between datasets, we did not report statistical significance (in terms of standard deviation). Instead, as we state in line 289 and 290: “for [the tested] 32B models, VC+TL is the best performing method in 110 out of 120 cases across three prompts, four reasoning models, and ten datasets”.
>
> We also run a bootstrap analysis with 1000 resampling runs, and report the standard deviations in Table 14. The standard deviations are usually within 1-2 points, and relatively stable. We expect the same to hold true throughout most experiments we present, since we use a modest sample size of 1K (relatively standard for LLM evaluations, and through the line of work we are following e.g., Yoon et al. 2025). We hope that this provides further statistical significance to our results, and are happy to run any further bootstraps and report them at the reviewer’s request.
>
> **W4**: “No Downstream Demonstration: The work stops at correlation analysis and does not explore how TL-based uncertainty estimation can improve real-world applications like answer filtering or self-consistency.”
>
> Our main finding is the identification of trace length as a useful and potentially interesting uncertainty signal. Uncertainty signals have a variety of uses (e.g., abstention), and we view a detailed study of these applications as being outside the scope of this work.
>
> We do, however, note that it is relatively straightforward to extend trace length to a measure which may be useful for practitioners of LLM UQ via fitting a trace length threshold to a hold-out set. We kindly refer to part 2 of our overall response for more discussion on this.

---

> > ### Author Response · Authors · 2025-11-22
> > **Response to questions**
> >
> > **Q1**: How stable is the correlation between trace length and uncertainty across different reasoning styles or prompt formats (e.g., Tree-of-Thought vs. Chain-of-Thought)?
> >
> > We evaluate trace length for four different prompts, and find it useful for each. We do not evaluate on more complex meta prompting strategies, such as Tree-of-Thought, and leave this to future work.
> >
> > **Q2**: Could trace length be confounded by model verbosity rather than genuine uncertainty?
> >
> > We do not know of a way to control for “pure” verbosity without affecting all aspects of an LLM’s responses. To the extent that a model chooses to be more verbose when it is uncertain, this is precisely the behavior we intend to capture using trace length.
> >
> > **Q3**: How would TL perform in creative or open-ended tasks where long outputs are not necessarily uncertain?
> >
> > In this work, we are explicitly looking at uncertainty about the correctness of the answer (rather than, for example, aleatoric variability in creative tasks). This makes it challenging to extend to open-ended tasks, where a response may not be fully "correct" or "incorrect". Yang et al. (2025) propose ways of evaluating uncertainty in this setting, by breaking down the problem into modular sub-tasks that can be correct or incorrect. In principle, trace length could be used in a similar pipeline; however it would require a robust method to divide the reasoning trace into modular subtasks.
> >
> > **Q4**: Can TL be combined with other uncertainty signals (e.g., entropy, logit variance) to build a more principled UQ framework?
> >
> > Indeed, in this work we show that TL can be combined with verbalized confidence. Recent work has indicated that there is indeed non-overlapping information among other (non-trace length) uncertainty signals, and that a better predictor can be obtained by combining these signals (Xiong et al. 2025). Finding a truly principled combination of these various signals is an open question for future work.
> >
> > Yang et al. (2025). LoGU: Long-form Generation with Uncertainty Expressions.
> >
> > Xiong et al. (2025). Efficient and Effective Uncertainty Quantification for LLMs.

---

### Official Review · Reviewer_9Lj7 · 2025-11-01

**Soundness:** 3
**Presentation:** 3
**Contribution:** 2
**Rating:** 4
**Confidence:** 3

**Summary:**

The paper investigate the mechanisms behind trace length’s performance as a confidence signal
The paper identify high-entropy or “forking” tokens as playing a key role in the mechanism.
Their findings demonstrate that reasoning post-training enhances uncertainty quantification beyond verbal expressions, and establish trace length as a practical confidence measure for large reasoning models.

**Strengths:**

- The paper aims to tackle an interesting problem, how to better perform zero-shot uncertainty estimation in reasoning models.
- The paper has presented extensive empirical results to verfiy trace length as a practical confidence measure for large reasoning models.

**Weaknesses:**

- In L149, the paper state that it aims to quantify “how various post-training approaches influence the verbalized confidence abilities of the resulting model.” However, it seems like all these reasoning models are tuned on the traces generated from R1, which raises questions about the generality of the findings.
- It is unclear to me how trace length is used in practice to judge the confidence of a single response. Does the method rely on a predefined threshold to decide whether a response is “confident” or “uncertain”? If so, how is this threshold determined and validated? Is it dataset dependent?
- It is unclear whether the conclusion hold across different RL post-training methods. Especially those where the reward function directly optimizes for reasoning trajectories.
- The paper shows that longer reasoning traces tend to correlate with less confidence answers, however, just based on the emperically experiments, It remains unclear whether trace length causes uncertainty or is simply a byproduct of other unknown confounders.

**Questions:**

see weakness

---

> ### Author Response · Authors · 2025-11-22
>
> We thank the reviewer for the feedback and careful consideration of our submission. We will address the weaknesses in order.
>
> **W1 and W3** Generalization across RL post-training methods: Please kindly refer to part 1 of our overall reviewer response.
>
> **W2** Actionable decision-making from trace length: Please kindly refer to part 2 of our overall reviewer response.
>
> **W4** Evidence for causal relationship between trace length and uncertainty: Indeed, we do not view our results as establishing a causal relationship between trace length and uncertainty. Instead, our focus is on zero-shot signals that are highly correlated with uncertainty and thus useful in practice. We do think the mechanism behind trace length’s correlation with uncertainty is an interesting question, and we explore both difficulty and forking tokens as potential contributors to this mechanism. We believe that forking tokens are a key component of the true underlying mechanism.

---

### Official Review · Reviewer_HZUW · 2025-11-10

**Soundness:** 4
**Presentation:** 4
**Contribution:** 3
**Rating:** 8
**Confidence:** 4

**Summary:**

This paper identifies trace length as a signal of a reasoning model's uncertainty. The authors show that trace length can be used as a signal zero-shot, provides complementary uncertainty signal to verbal uncertainty, is not accounted for by other correlates like problem difficulty, and seems to be attributed to the presence of "forking tokens" like "wait" and "maybe."

**Strengths:**

- Interesting headline result: trace length correlates with uncertainty, and in a way that complements verbal confidence
- Thorough, comprehensive experiments: controlling for question difficulty, comparison to verbal confidence. Contributes to the community's understanding of reasoning models and uncertainty.
- Clear and well-written paper

**Weaknesses:**

I believe this is a strong scientific result with no weaknesses that impact my recommendation of acceptance. There are limitations of the method, most of which the authors point out:
- limited generalizability, particularly in low-accuracy regimes where it is particularly important to have uncertainty estimation
- trace length might only be a reliable (correlational) signal given current model training pipelines, rather than a fundamental property of model reasoning. Specifically:
	- if we optimize models for other length-related properties or behaviors that impact length (e.g. exploration), trace length probably wouldn't be a reliable signal of uncertainty any more -- or at least, it would be unclear how the result in this paper generalizes.
	- while the paper identifies "forking tokens" as a "mechanism" for the correlation between trace length and uncertainty, the importance of tokens like "wait" or "maybe" being forking tokens seems like an artifact of current model training. The authors argue that that forking tokens can more generally be defined as any token where the LLM has "high entropy in its token distribution," but then this seems like a vacuous definition of uncertainty.

**Questions:**

- It seems important to see the procedure for how trace length is used to compute AUROC explicitly written down in the main body of the paper. For verbalized and numerical confidence, we know the range of bins -- for trace length, do you normalize by the min/max trace length for a particular model/dataset combination?
- Related to above: how much does the normalization across datasets impact the reliability of trace length as a signal? The bins for verbalized confidence seem to work regardless of what dataset we apply them to, but I'm wondering how we should interpret the model's confidence on dataset A where reasoning traces are between 10-100 tokens vs. dataset B where the min/max trace lengths are 50-10000 tokens.

---

> ### Author Response · Authors · 2025-11-22
>
> We sincerely thank the reviewer for the positive comments and feedback on our work. We agree with most of the stated weaknesses of the method, and provide slightly more context for each.
>
> **W1** Limited utility in low-accuracy regimes:  We agree that in general zero-shot methods are unlikely to perform well in low accuracy regimes. We emphasize that even VC becomes less useful here as well (see figure 18 in Appendix K, and also Mei et al. (2025)).
>
> **W2** Dependence on Training Pipelines: We agree that trace length (TL) may only be a reliable signal given current model training pipelines. We will, however, point out that TL as a signal does seem to be robust to a variety of current pipelines. We kindly refer to our new experiments in Appendix L (and our overall reviewer response). However, since the algorithms and recipes for many of these models post-training pipelines are not public, it is unknown whether they try to control for generation length.
>
> **Q1**: Please refer to part 2 of our overall response.
>
> **Q2**: Thank you for the insightful question. Indeed, trace length does not naturally fall into a predefined scale (in particular, it cannot be interpreted as a probability even after dividing by the max trace length) and may be difficult to compare across model / dataset combinations. We view this as being inherent to the problem as different model / dataset combinations necessarily have different response behaviors, and trace length must always be interpreted in the context of a particular model / dataset. (If it is necessary to compare trace lengths in different settings, we believe a good way to do so is it to interpret trace lengths in terms of percentiles on their respective distributions.) We have added additional discussion of this behavior and recommendations for maximizing discriminative ability in Appendix B, lines 1087-1095.

---

### Author Response · Authors · 2025-11-22
**Response to Common Reviewer Questions**

We thank all reviewers for their helpful comments and suggestions to improve our work. Note that we have updated the PDF draft, and all new content / edits are marked in a magenta color. We have added two additional appendix sections (L and M), and address some common concerns and questions here.

**Part 1**: The most common concern among reviewers was that our experiments are limited in scope and run only on reasoning models derived from applying SFT, RL, or SFT+RL on Qwen2.5 base models with R1 reasoning traces. We prioritized these models as they are open-weight models for which the post-training procedure is transparent and publicly available.

To establish the generality of our results, we have introduced new experiments in Appendix L on four new reasoning models: Phi-4-reasoning-plus-14B, Hermes-70B, GPT-OSS-20B, and GLM 4.5 AIR-106. These models were trained with multiple different post-training procedures, the details of which are not fully public (for example, GPT-OSS appears to have been trained explicitly to minimize verbosity). We observe that trace length remains a useful indicator for uncertainty and performs comparably with verbalized confidence. Furthermore, combining trace length and verbal confidence also performs favorably.

In summary, our newly added experiments in Appendix L plus experiments from our main paper address generalization to other architectures, models, and post-training algorithms.

**Part 2**: Another common question was: how do we compute AUROC of trace length? Furthermore, how do we turn trace length into an actionable uncertainty signal?

We clarify that computing the ROC curve requires only a calibration dataset of the form $\{(x_i, y_i, s_i)\}_{i}$, where $(x_i, y_i)$ is the $i$th labeled example, and $s_i \in \mathbb{R}$ an associated (unnormalized) ``uncertainty score'' (for instance, the numeric verbal confidence or the trace length). We have added additional discussion to clarify this in the main paper (lines 197-200, 214-15).

There are a variety of ways that practitioners might make use of a scalar uncertainty score like trace length. Fundamentally, the ROC summarizes the utility of taking actions by thresholding the underlying uncertainty score. The most common and representative approach is abstention: “if score > threshold, abstain”. This threshold must be determined in a context-specific way (e.g., separately for each model / dataset combination), balancing true positives against false positives as desired according to the ROC. Higher area under the ROC (AUROC) generally indicates better tradeoffs between these two quantities and thus a more desirable uncertainty metric. Once the desired TPR/FPR balance is determined by a practitioner, the threshold can be calculated using a holdout validation set, optionally with conformal methods for rigorous guarantees.

---

### Meta-Review · Area_Chair_LgP3 · 2026-01-09

**Summary:**

The submission uses empirical evidence to argue that reasoning-trace length is a simple uncertainty signal for reasoning-tuned LLMs, often comparable to verbalized confidence and sometimes improved when combined with it. Reviewers generally agree the empirical correlation is real and the paper is clearly written. However, a consistent concern is with the depth of the insight, and the usefulness of it. The paper's analyses are limited to a correlation study, providing little machanisitc understanding (the forking-token argument etc. is still based on a correlation study). The boost in the accuracy of verbalized confidence is also very marginal.

On balance, I do not recommend acceptance. I encourage the authors to strengthen the paper by (i) positioning and benchmarking against stronger UQ baselines, (ii) adding a concrete downstream use case where TL improves outcomes, and (iii) deepening the mechanistic analysis and providing clearer conditions under which TL is expected to remain reliable.

**Reviewer Concerns:**

The core insight offers limited novelty: the main contribution is broad validation of a straightforward heuristic rather than a new method, principle, or theory. Several reviewers also found the mechanistic analysis (e.g., “forking/high-entropy tokens”) suggestive but not fully explanatory, and questioned whether the effect is robust beyond current reasoning post-training pipelines (especially if future training objectives explicitly regularize verbosity/length). Reviewers also question the usefulness of this insight.

The rebuttal addresses some factual/methodological issues and adds breadth, but it does not resolve the central issue that the contribution is incremental and potentially pipeline-dependent.

**Reviewer Scores:**

I think one reviewer may raise their score from 4 to 6. But the other two reviewers are still not supportive of the acceptance.

---

### Decision · Program_Chairs · 2026-01-26

Reject